# What Breaks the Curse of Dimensionality in Deep Learning?

## Abstract

Although learning in high dimensions is commonly believed to suffer from the curse of dimensionality, modern machine learning methods often exhibit an astonishing power to tackle a wide range of challenging real-world learning problems without using abundant amounts of data. How exactly these methods break this curse remains a fundamental open question in the theory of deep learning. While previous efforts have investigated this question by studying the data (D), model (M), and inference algorithm (I) as independent modules, in this paper we analyzes the triple (D, M, I) as an integrated system. We examine the basic symmetries of such systems, focusing on four of the main architectures in deep learning: fully-connected networks (FCN), locally-connected networks (LCN), and convolutional networks with and without pooling (GAP/VEC). By computing an eigen-decomposition of the infinite-width limits (aka Neural Kernels) of these architectures, we characterize how inductive biases (locality, weight-sharing, pooling, etc) and the breaking of spurious symmetries can affect the performance of these learning systems. Our theoretical analysis shows that for many real-world tasks it is locality rather than symmetry that provides the first-order remedy to the curse of dimensionality. Empirical results on state-of-the-art models on ImageNet corroborate our results.

## 1 Introduction

Statistical problems with high-dimensional data are frequently plagued by the *curse of dimensionality*, in which the number of samples required to solve the problem grows rapidly with the dimensionality of the input. Classical theory explains this phenomenon as the consequence of basic geometric and algebraic properties of high-dimensional spaces; for example, the number of $\epsilon$-cubes inside a unit cube in $\mathbb{R}^d$ grows exponentially like $\epsilon^{-d}$, and the number of degree $r$ polynomials in $\mathbb{R}^d$ grows like a power-law $d^r$. Since for real-world problems $d$ is typically in the hundreds or thousands, classical wisdom suggests that learning is likely to be infeasible. However, starting from the groundbreaking work AlexNet [1], practitioners in deep learning have tackled a wide range of difficult real-world learning problems ([2–6]) in high dimensions, once believed by many to be out-of-scope of current techniques. The astonishing success of modern machine learning methods clearly contradicts the curse of dimensinonality and therefore poses the fundamental question: mathematically, how do modern machine learning methods break the curse of dimensionality?

To answer this question, we must trace back to the most fundamental ingredients of machine learning methods. They are the data ($\mathcal{D}$), the model ($\mathcal{M}$), and the inference algorithm ($\mathcal{I}$).

Data ($\mathcal{D}$) is of course central in machine learning. In the classical learning theory setting, the learning objective usually has a power-law decay $m^{-\beta}$ as the function of the number of training samples $m$. The theoretical bound on $\beta$ is usually tiny, owing to the curse of dimensionality, and is of

limited practical utility for high-dimensional data. On the other hand, empirical measurements of $\beta$ in state-of-the-art deep learning models typically reveal values of $\beta$ that are not at all small (e.g. $\beta = 0.43$ for ResNet in Fig.S2) even though $d$ is quite large (e.g. $d \sim 10^5$ for ImageNet). This example suggests that the learning curve must have important functional dependence on $\mathcal{M}$ and $\mathcal{I}$. Indeed, as we will observe later, many of the best performing methods exhibit learning curves for which $\beta = \beta(m)$ actually *increases* as $m$ becomes larger, i.e. data makes the usage of data more efficient. We call this phenomenon DIDE, for **d**ata **i**mproves **d**ata **e**fficiency.

Designing machine learning models ($\mathcal{M}$) that maximize data-efficiency is critical to the success of solving real-world tasks. Indeed, breakthroughs in machine learning are often driven by novel architectures LeNet [7], AlexNet[1], Transformer [2], etc. While some of the inductive biases of these methods are clear (e.g. translation symmetries of CNNs), others tend to build off of prior empirical success and are less well-understood (e.g. the implicit bias of SGD). To build our understanding of these biases and how they affect learning, we conduct a theoretical analysis of them in the infinite-width setting [8–12], which preserves most salient aspects of the architecture while enabling tractable calculations. We classify all phenomena that could be explained by infinite networks alone as the consequences of inductive biases.

The inference procedure ($\mathcal{I}$) is what enables *learning* in machine learning methods. It is widely believed that modern inference methods, specifically gradient descent and variants, 'implicitly' bias the solutions of the networks towards those that generalize well and away from those that generalize poorly [13–15]. The effects of the inference algorithm are intimately tied to the specifics of the model (e.g. weight-sharing) and the data (e.g. augmentation), and might not be fully understood with a fixed-data, fixed-model analysis. Indeed, good performance may derive from interactions between $(\mathcal{M}, \mathcal{I})$, or $(\mathcal{D}, \mathcal{I})$, or even $(\mathcal{D}, \mathcal{M}, \mathcal{I})$. In Sec. 3.1, we demonstrate the DIDE effect for a particular choice of $(\mathcal{D}, \mathcal{M}, \mathcal{I})$ and show that this effect disappears if any one of $\mathcal{D}$, $\mathcal{M}$, or $\mathcal{I}$ is altered.

The above discussion highlights the insufficiency of treating $\mathcal{D}$, $\mathcal{M}$, and $\mathcal{I}$ as separate non-interacting modules. They must be considered as an integrated system. Throughout this paper, we will refer to the triplet $(\mathcal{D}, \mathcal{M}, \mathcal{I})$ as a (machine) learning system and the tuple $(\mathcal{M}, \mathcal{I})$ as the learning algorithm of the system that operates on $\mathcal{D}$. We summarize our contributions below.

1. We surface the basic symmetries of various $(\mathcal{D}, \mathcal{M}, \mathcal{I})$associated to four of the main architectures in deep learning $\mathsf{FCN}_n$ (fully-connected networks), $\mathsf{LCN}_n$ (locally-connected networks), $\mathsf{VEC}_n/\mathsf{GAP}_n$ (convolution networks with a flattening /a global average pooling readout layer), their infinite width counterparts $\mathsf{FCN}_\infty/\mathsf{LCN}_\infty/\mathsf{VEC}_\infty/\mathsf{GAP}_\infty$. Treating $\mathsf{FCN}_{n/\infty}$ as the baseline model, we show that the locality from $\mathsf{LCN}_n$ and the weight-sharing from $\mathsf{VEC}_n/\mathsf{GAP}_n$ break spurious symmetries and lead to better systems. Empirically, we examine the relation between the symmetries and the performance of the systems in the infinite width setting and finite width setting with various of interventions. Surprisingly, we observe that state-of-the-art learning system (EfficientNet[16]) on ImageNet can learn almost equally well even the coordinate of the data are transformed by the symmetry group defined by $\mathsf{LCN}_n$.

2. We show that although the weight-sharing from $\mathsf{VEC}_n$ provides coordinate information of the data to the system, as the width gets larger, it becomes harder for the learning algorithm to explore such information and at infinite width, the system restores the symmetry group that is identical to $\mathsf{LCN}_n$, and is completely unaware of the coordinate information. As a consequence, the performance of the network, as a function of width, monotonically decays [12]. This is in stark contrast to recent finding that the performance of network is positively correlated to its width. We show that this phenomenon continues to hold even with various interventions (larger learning rate and l2 regularization) to the training procedures. However, with more data (e.g. data augmentation) $\mathsf{VEC}_n$ can be on par with $\mathsf{GAP}_n$.

3. The function space defined by $\mathsf{LCN}_n$ is a super set of that defined by $\mathsf{VEC}_n$. We prove the opposite is true. Therefore, $\mathsf{VEC}_n$ is able to express functions in the space with a stronger inductive bias $\mathsf{GAP}_n$ (translation invariance) and functions in a seemingly much larger class $\mathsf{LCN}_n$. We hypothesize that as the dataset grows, the learned functions using $\mathsf{VEC}_n$ is transitioned away from those learned using $\mathsf{LCN}_n$ and become closer to those learned using $\mathsf{GAP}_n$. This suggests, even though the prior (provided by human) is not 100% correct, with the help of more data, gradient descent might be able to correct it, a possible explanation of DIDE.

4. When the input space is the product of hyperspheres, we eigendecompose the kernels associated to one-hidden layer infinite width network, $\mathsf{FCN}_\infty$, $\mathsf{VEC}_\infty = \mathsf{LCN}_\infty$ and $\mathsf{GAP}_\infty$. We treat $\mathsf{FCN}_\infty$ as the baseline, whose order $r$ eigenspace has dimension of order $d^r$ and eigenvalues of order $d^{-r}$ for $r \geq 0$ [17]. We show that locality alone (i.e. $\mathsf{VEC}_\infty$) dramatically reduces the dimension of the $r$-eigenspace for $r \geq 2$ and the spectral gap between all $r$-eigenspaces but $r = 0$ and $r = 1$, making learning of higher order eigenspaces feasible with dramatically fewer samples and gradient steps. In addition, pooling (i.e. $\mathsf{GAP}_\infty$) reduces the dimension of $r$-eigenspace for $r \geq 1$ by a factor equal to the size of the pooling window, but it does not change the spectra in an essential way.

Our empirical and theoretical results surface the importance of locality which, we believe, provides the first-order remedy to the curse of dimensionality for many real-world tasks and which has been largely overlooked.

# 2 Preliminary and Notation

## 2.1 Neural Networks

We focus our presentation on the supervised learning setting and more concretely, on image recognition. Let $\mathcal{D} \subseteq (\mathbb{R}^d)^3 \times \mathbb{R}^k \equiv \mathbb{R}^{3d} \times \mathbb{R}^k$ denote the data set (training and test) and $\mathcal{X} = \{x : (x, y) \in \mathcal{D}\}$ and $\mathcal{Y} = \{y : (x, y) \in \mathcal{D}\}$ denote the input space (images) and label space, respectively. Here $d$ is the spatial dimension (e.g. $d = 32 \times 32$ for CIFAR-10) of the images and 3 is the total number of channels (i.e. RGB). We use $\mathsf{FCN}_n$ to denote a $L$-hidden layer fully-connected network with identical hidden widths $n_l = n \in \mathbb{N}$ for $l = 1, ..., L$ and with readout width $n_{L+1} = k$ (the number of logits). For each $x \in \mathbb{R}^{3d} = (\mathbb{R}^d)^3$, we use $h^l(x), x^l(x) \in \mathbb{R}^{n_l}$ to represent the pre- and post-activation functions at layer $l$ with input $x$. The recurrence relation FCN is given by

$$\begin{cases} h^{l+1} & = x^l W^{l+1} \\ x^{l+1} & = \phi\left(h^{l+1}\right) \end{cases} \text{ and } W_{i,j}^l = \frac{1}{\sqrt{n_l}} \omega_{ij}^l, \quad \omega_{ij}^l \sim \mathcal{N}(0, 1) \tag{1}$$

where $\phi$ is a point-wise activation function, $W^{l+1} \in \mathbb{R}^{n_l \times n_{l+1}}$ are the weights and $\omega_{ij}^l$ are the trainable parameters, drawn i.i.d. from a standard Gaussian $\sim \mathcal{N}(0, 1)$ at initialization. For simplicity of the presentation, the bias terms and the hyperparameters (the variances of the weights) are omitted. Adding them back won't affect the conclusion of the paper.

For convolutional networks or locally-connected networks, the inputs are treated as tensors in $(\mathbb{R}^d)^3$. The recurrent relation of convolutional networks can be written as

$$x_{\alpha,j}^{l+1} = \phi(h_{\alpha,j}^{l+1}) \quad \text{and} \quad h_{\alpha,j}^{l+1} \equiv \frac{1}{\sqrt{(2k+1)n^l}} \sum_{j=1}^{n^l} \sum_{\beta=-k}^{k} x_{\alpha+\beta,i}^l \omega_{ij,\beta}^l \tag{2}$$

Here $\alpha \in [d]$ denote the spatial location, $i/j \in [n]$ denotes the fanin/fanout channel indices. For notational convenience, we assume circular padding and stride equal to 1 for all layers. The features of the penultimate layer are 2D tensors and there are two commonly used approaches to map them to the logit layer: stack a dense layer after either vectorizing the 2D tensor to a 1D vector or applying a global average pooling layer to each channel. We use $\mathsf{VEC}_n/\mathsf{GAP}_n$ to denote the network obtain from the former/latter, which are known to be equipped with the inductive biases translation equivariant/invariant. The readout layer of $\mathsf{VEC}_n/\mathsf{GAP}_n$ could be written as

$$x_j^{L+1} = \frac{1}{\sqrt{dn}} \sum_{\alpha \in [d]} x_{\alpha,i}^L w_{\alpha,ij}^{L+1}, \quad x_j^{L+1} = \frac{1}{\sqrt{n}} \sum_{i \in [n]} \left( \frac{1}{d} \sum_{\alpha \in [d]} x_{\alpha,i}^L \right) w_{ij}^{L+1} \tag{3}$$

We briefly remark the the key difference between the two. In $\mathsf{VEC}_n$, each pixel in the penultimate layer has its own (independent random) variable while pixels within the same channel shared the same (random) variable in $\mathsf{GAP}_n$. It is clear that the function space of $\mathsf{VEC}_n$ contains that of $\mathsf{GAP}_n$. Locally Connected Networks $\mathsf{LCN}_n$ [18, 19] are convolutional network without weight sharing between spatial locations. $\mathsf{LCN}_n$ preserve the connectivity pattern, and thus topology, of a convnet. Mathematically, the current formula is defind as in Equation 2 with all the *shared* parameters $\omega_{ij,\beta}^l$ replaced by *unshared* $\omega_{ij,\alpha,\beta}^l \sim \mathcal{N}(0, 1)$

In this note, we assume that the $\mathsf{LCN}_n$ are always associated with a vectorization readout layer and it is clear, as a function space, $\mathsf{LCN}_n$ is a super set of $\mathsf{VEC}_n$. Interestingly,the opposite is also true.

**Theorem 2.1** (Sec. B). *Let* $\mathsf{VEC}_n/\mathsf{LCN}_n/\mathsf{GAP}_n$ *denote the set of functions that can be represented by L-hidden layer* $\mathsf{VEC}_n/\mathsf{LCN}_n/\mathsf{GAP}_n$ *networks with hidden width n. Then*

$$\mathsf{GAP}_n \subseteq \mathsf{VEC}_n \subseteq \mathsf{LCN}_n \subseteq \mathsf{VEC}_{dn} \tag{4}$$

The significance of this theorem is that if we consider the function space $\mathsf{VEC}_n$ as a soft *prior*, gradient descent could move it closer to a *better* prior $\mathsf{GAP}_n$ (translation invariance) if the average pooling is (approximately) learned in the readout layer or it might remain close to $\mathsf{LCN}_n$.

## 2.2 Gradient Descent Training

We use $f$ to denote any functions defined by the architectures above and $\theta$ to denote the collection of all parameters. Denote by $\theta_t$ the time-dependence of the parameters and by $\theta_0$ their initial values. We use $f_t(x) \equiv f(x, \theta_t) \in \mathbb{R}^k$ to denote the output (or logits) of the neural network at time $t$. Let $\ell(\hat{y}, y) : \mathbb{R}^k \times \mathbb{R}^k \to \mathbb{R}$ denote the loss function where the first/second argument is the prediction/true label. By applying continuous time gradient descent to minimize the objective $\mathcal{L} = \sum_{(x,y)\in\mathcal{D}} \ell(f_t(x, \theta), y)$, the evolution of the parameters $\theta$ and the logits $f$ can be written as

$$\dot{\theta}_t = -\nabla_\theta f_t(\mathcal{X}_T)^T \nabla_{f_t(\mathcal{X}_T)}\mathcal{L}, \qquad \dot{f}_t(\mathcal{X}_T) = \nabla_\theta f_t(\mathcal{X}_T)\dot{\theta}_t = -\hat{\Theta}_t(\mathcal{X}_T, \mathcal{X}_T)\nabla_{f_t(\mathcal{X}_T)}\mathcal{L} \tag{5}$$

where $f_t(\mathcal{X}_T) = \mathrm{vec}\left([f_t(x)]_{x\in\mathcal{X}_T}\right)$, the $k|\mathcal{D}| \times 1$ vector of concatenated logits for all examples, and $\nabla_{f_t(\mathcal{X}_T)}\mathcal{L}$ is the gradient of the loss with respect to the model's output, $f_t(\mathcal{X}_T)$. $\hat{\Theta}_t \equiv \hat{\Theta}_t(\mathcal{X}_T, \mathcal{X}_T)$ is the tangent kernel at time $t$, which is a $k|\mathcal{D}| \times k|\mathcal{D}|$ kernel matrix

$$\hat{\Theta}_t = \nabla_\theta f_t(\mathcal{X}_T)\nabla_\theta f_t(\mathcal{X}_T)^T \tag{6}$$

One can define the tangent kernel for general arguments, e.g. $\hat{\Theta}_t(x, \mathcal{X}_T)$ where $x$ is test input. At finite-width, $\hat{\Theta}$ will depend on the specific random draw of the parameters and evolve with time. As such, for a test point $x$ the prediction $f_t(x)$ depends on the random initalization and is also stochastic. Note that the parameters are initialized randomly and the randomness will be carried out through the training procedure. As a consequence, the prediction functions are stochastic.

## 2.3 Infinite Network: Gaussian Processes and the Neural Tangent Kernels

**Neural Networks as Gaussian Processes (NNGP).** As the width $n \to \infty$, at initialization the output $f_0(\mathcal{X})$ forms a Gaussian Process $f_0(\mathcal{X}) \sim \mathcal{GP}(0, \mathcal{K}(\mathcal{X}, \mathcal{X}))$, known as the NNGP [8, 20, 21]. Here $\mathcal{K}$ is the GP kernel and can be computed in closed form for a variety of architectures. By treating this infinite width network as a Bayesian model (aka Bayesian Neural Networks) and applying Bayesian inference, the posterior is also a GP

$$\mathcal{N}\left(\mathcal{K}(\mathcal{X}_*, \mathcal{X}_T)\mathcal{K}^{-1}(\mathcal{X}_T, \mathcal{X}_T)\mathcal{Y}_T, \mathcal{K}(\mathcal{X}_*, \mathcal{X}_*) - \mathcal{K}(\mathcal{X}_*, \mathcal{X})\mathcal{K}(\mathcal{X}, \mathcal{X})^{-1}\mathcal{K}(\mathcal{X}_*, \mathcal{X})^T\right) \tag{7}$$

**Neural Tangent Kernelss(NTK).** Recent advance in global convergence theory of overparameterized networks [22–25, 12] has shown that under certain assumptions, the tangent kernels is almost stationary over the course of training and is concentrated on its infinite width limit $\Theta$ in the sense there is a constant $C$ independent of $t$ and the network's width $n$ such that

$$\sup_{t\geq 0}\|\hat{\Theta}_t(\mathcal{X}_T, \mathcal{X}_T) - \Theta(\mathcal{X}_T, \mathcal{X}_T)\|_F + \|\hat{\Theta}_t(\mathcal{X}_T, \mathcal{X}_*) - \Theta(\mathcal{X}_T, \mathcal{X}_*)\|_F \leq \frac{C}{\sqrt{n}}. \tag{8}$$

where is the infinite width limit of $\Theta$ at initialization, whose existence has been proved in [22, 26]. As such, when the loss is the mean squared error (MSE), the mean prediction (margininated over random initialization) has the following closed form

$$f(\mathcal{X}_*) = \Theta\left(\mathcal{X}_*, \mathcal{X}_T\right)\Theta^{-1}(\mathcal{X}_T, \mathcal{X}_T)\left(I - e^{-\eta\Theta(\mathcal{X}_T, \mathcal{X}_T)t}\right)\mathcal{Y}, \tag{9}$$

Letting $t \to \infty$, the above solution is the same as that of the kernel ridgeless regression using the infinite width tangent kernel $\Theta$. We use $\mathsf{FCN}_\infty(x)$, $\mathsf{LCN}_\infty(x)$, $\mathsf{VEC}_\infty(x)$ and $\mathsf{GAP}_\infty(x)$ to denote the infinite width solutions (either the GP inference or the NTK regression) for the corresponding architectures, where we have suppressed the dependence on the training data $(\mathcal{X}_T, \mathcal{Y}_T)$.

## 3 Symmetries of Machine Learning Systems

Symmetry is fundamental in physical systems. So is it in machine learning systems. We explore symmetries of various machine learning systems in this section. Given $\mathcal{D} = (\mathcal{X}, \mathcal{Y})$ and a transformation on the input space $\tau : \mathbb{R}^{3d} \to \mathbb{R}^{3d}$, we set $\tau(\mathcal{D}) = (\tau(\mathcal{X}), \mathcal{Y})$. Let $O(3d)$ denote the orthogonal group on the flatten input space $\mathbb{R}^{3d}$. The subgroup $O(3)^d \leq O(3d)$ operates on the un-flattened input $(\mathbb{R}^d)^3$, whose element rotates each pixel $x_\alpha \in \mathbb{R}^3$ by an independent element $\tau_\alpha \in O(3)$. The smaller subgroup $O(3) \otimes \mathbf{I}_d \leq O(3)^d$ applies the *shared* rotation (i.e. $\tau_\alpha = \tau$ to all $x_\alpha$ for $\alpha \in [d]$). We use $P(3d)$ to denote the permutation group on $\mathbb{R}^{3d}$ and $P(3)^d$ and $P(3) \otimes \mathbf{I}_d$ are defined similarly. Note that rotating $\mathcal{X}$ by $\tau$ is equivalent to transfer the original coordinate system by the adjoint tranformation $\tau^* = \tau^{-1}$.

For a deterministic (stochastic) learning algorithm $\mathcal{A} = (\mathcal{M}, \mathcal{I})$, we use $\mathcal{A}(\mathcal{D}_T)$ to denote the learned function (distribution of the learned functions) using training set $\mathcal{D}_T$. We use $\mathcal{A}^\tau(\mathcal{D}_T)$ to denote the learned function(s) using $\tau(\mathcal{D}_T)$ and makes prediction on the transformed test point $\tau(\mathcal{X}_*)$. In another word, the learning algorithm is conducted in the input space whose coordinate system is transformed by $\tau^{-1}$.

**Definition 1.** *Let $\mathcal{G}$ be a group of transformations $\mathbb{R}^{3d} \to \mathbb{R}^{3d}$. We say a deterministic (stochastic) learning algorithm $\mathcal{A} = (\mathcal{M}, \mathcal{I})$ is g-invariant if $\mathcal{A} = \mathcal{A}^g$ ($\mathcal{A} =^d \mathcal{A}^g$ ). In this case, we say the system $(\mathcal{D}, \mathcal{M}, \mathcal{I})$ is g-invariant and use the notation $(\mathcal{D}, \mathcal{M}, \mathcal{I}) = (g\mathcal{D}, \mathcal{M}, \mathcal{I})$. If this holds for all $g \in \mathcal{G}$, then we say the algorithm and the system are $\mathcal{G}$-invariant.*

If $(\mathcal{M}, \mathcal{I})$ is the algorithm of minimum norm linear regressor, then $(\mathcal{D}, \mathcal{M}, \mathcal{I})$ is $O(3)^d$-invariant; see Sec.G for more details. Note that the symmetry (invariance) in our definition is a property of a system and is different from the notion of symmetry that are commonly used in the machine learning community, which is a property of a function (e.g. translation invariance).

**Theorem 3.1** (Sec.C). *If the parameters of the networks are initialized with iid $\mathcal{N}(0, 1)$, then*

- $\mathsf{FCN}_{n/\infty}$ are $O(3d)$-invariant.
- $\mathsf{LCN}_{n/\infty}$ are $O(3)^d$-invariant.
- $\mathsf{VEC}_n$ is $O(3) \otimes \mathbf{I}_d$-invariant and $\mathsf{VEC}_\infty$ is $O(3)^d$-invariant.
- $\mathsf{GAP}_{n/\infty}$ are $O(3) \otimes \mathbf{I}_d$-invariant.

The $O(3d)$-invariant of $\mathsf{FCN}_\infty$ is because the NTK/NNGP kernel is an inner product kernel, namely, there is a function $k$ such that the kernels have the form $k(\langle x, x' \rangle)$. The $O(3d)$-invariant of finite width $\mathsf{FCN}_n$ is due to the Gaussian initialization of the first layer which was first observed and proved in [27]. Rotating the input by $\tau \in O(3d)$ is equivalent to rotating the weight matrix $\omega$ of the first layer by $\tau^*$. Since for $\omega \in \mathcal{N}(0, 1)^{3d}$ $\tau^*\omega =^d \omega$, at random initialization, the distribution of the output functions (the prior) are unchanged if all inputs are rotated by the same element in $O(3d)$. This property continues to hold throughout the course of (continue/discrete) gradient descent training with/without $L^2$-regularization and Bayesian posterior inference. For the same reason, $\mathsf{LCN}_n$ is $O(3)^d$-invariant because each patch of the image uses independent Gaussian random variables. However, weight-sharing in $\mathsf{VEC}_n$ and $\mathsf{GAP}_n$ breaks the $O(3)^d$ symmetry, reducing it to $O(3) \otimes \mathbf{I}_d$.

For infinite networks, $\mathsf{LCN}_\infty = \mathsf{VEC}_\infty$ [28–31]. The kernels of $\mathsf{VEC}_\infty$ and $\mathsf{GAP}_\infty$ are of the forms

$$\Theta_{\mathsf{VEC}}(x, x') = k(\{\langle x_\alpha, x'_\alpha \rangle\}_{\alpha \in [d]}) \quad \text{and} \quad \Theta_{\mathsf{GAP}}(x, x') = k(\{\langle x_\alpha, x'_{\alpha'} \rangle\}_{\alpha, \alpha' \in [d]}), \tag{10}$$

resp. The former depends only on the inner product between pixels in the same spatial location, breaking the $O(3d)$ symmetry and reducing it to $O(3)^d$. In addition, the latter depends also on the inner products of pixels across different spatial locations due to pooling, which breaks the $O(3)^d$ symmetry and reduces it to $O(3) \otimes \mathbf{I}_d$.

Note that $\dim(O(3d)) = 3d(3d-1)/2$, $\dim(O(3)^d) = 3d$ and $\dim(O(3) \otimes \mathbf{I}_d) = 3$. $\mathsf{LCN}_n/\mathsf{VEC}_\infty$ dramatically reduces the dimension of the symmetry group. It is worth mentioning that while $\dim(O(3d))$ many pairs of rotated and unrotated images are needed to recover the exact rotation in $O(3d)$, only 3 pairs are sufficient for $O(3)^d$, same as that of $O(3) \otimes \mathbf{I}_d$. The results of the paper are presented in the most *vanilla* setting. Our methods can easily extend to more complicated architectures like ResNet[32], MLP-Mixer[33] and etc. The symmetry groups of such systems need to be computed in a case-by-case manner by identifying the invariant group of the random initialization and training procedures.

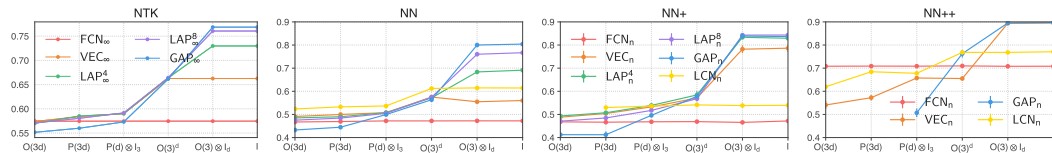

Figure 1: **Performance vs Symmetry**. Machine learning systems are equipped with various kinds of symmetries. Transforming the system by the associated symmetry does not affect the performance of the system. However, injecting spurious symmetries beyond the associated symmetries could dramatically degrade their performance for both finite and infinite networks.

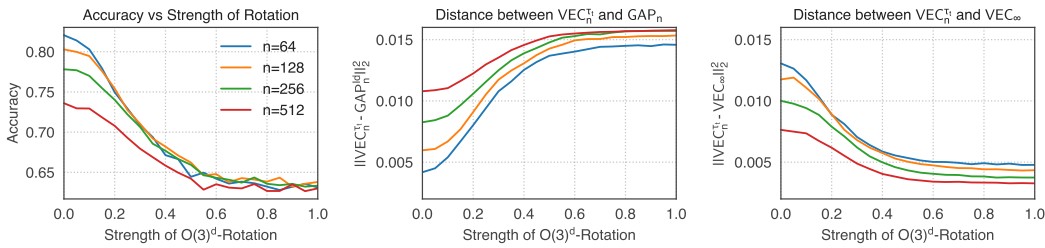

Figure 2: Even in the NN+ setting, $\mathsf{VEC}_n$ is closer to $\mathsf{GAP}_n$ for small $n$ and moves towards $\mathsf{VEC}_\infty$ with more symmetries and/or larger $n$ and accuracy drops.

### 3.1 Empirical Supports and Observations

**Performance under Rotations.** We examine the performance of: $\mathsf{FCN}, \mathsf{VEC}, \mathsf{LCN}, \mathsf{GAP}$ and $\mathsf{LAP}^{4/8}$, when the coordinates of the data are transformed by six different groups ($x$-axis in Fig.1) using the standard dataset CIFAR-10. , Here $\mathsf{LAP}^{4/8}$ is the same as $\mathsf{GAP}$ except the readout layer is replaced by the **L**ocal **A**verage **P**ooling with window size $4 \times 4/8 \times 8$. We consider 4 types of training methods: (1) $\mathsf{NTK}$, i.e. infinite networks (2)NN, our baseline for finite width neural network which is trained with momemtum using a small learning rate and without $L^2$ regularizer and the network is centered ($+\mathsf{C}$) to reduce the variance from random initialization (3)NN+= NN+LR+L2−C, i.e. using a larger learning rate ($+\mathsf{LR}$), adding $L^2$ regularization ($+\mathsf{L2}$)) and removing the centering ($-\mathsf{C}$) (4) NN++=NN++DA, adding MixUp[34] data augmentation ($+\mathsf{DA}$) to NN+. Overall, we observe that, for most of the cases in NTK/NN/NN+, adding spurious symmetry to a system $(\mathcal{D}, \mathcal{M}, \mathcal{I})$ degrades the performance towards that of the system invariant to that symmetry. Surprisingly, in the baseline NN, performance of $\mathsf{VEC}_n + \mathrm{O}(3) \otimes \mathbf{I}_d$ rotation is slightly worse than that of $\mathsf{VEC}_n + \mathrm{O}(3)^d$ and than that of $\mathsf{LCN}_n$, indicating that the system with $\mathcal{M} = \mathsf{VEC}_n$ is likely operating closely on the $\mathrm{O}(3)^d$ symmetry. The interventions $-\mathsf{C}+\mathsf{L2}+\mathsf{LR}$ in NN+ distinguishes the performance of $\mathsf{VEC}_n + \mathrm{O}(3) \otimes \mathbf{I}_d$ from $\mathsf{VEC}_n + \mathrm{O}(3)^d$ and $+\mathsf{DA}$ eventually closes the performance gap between $\mathsf{VEC}_n + \mathrm{O}(3) \otimes \mathbf{I}_d$ and $\mathsf{GAP}_n + \mathrm{O}(3) \otimes \mathbf{I}_d$, helping the system to be aware of the smaller symmetry $\mathrm{O}(3) \otimes \mathbf{I}_d$, escaping from the $\mathrm{O}(3)^d$ symmetry.

**Symmetry Breaking of $\mathsf{VEC}_n$.** Assuming Equation 8, namely, the network is in the NTK regime,

$$\lim_{n \to \infty} |\mathbb{E}\mathsf{VEC}_n(x) - \mathsf{VEC}_\infty(x)| + \lim_{n \to \infty} |\mathbb{E}\mathsf{VEC}_n(x) - \mathbb{E}\mathsf{VEC}_n^\tau(x)| \le Cn^{-\frac{1}{2}} \tag{11}$$

where the expectation $\mathbb{E}$ is over random initialization and $\mathsf{VEC}_n(x)$ is the prediction of the test point $x$ when $t = \infty$, i.e. training loss is 0. $\mathsf{VEC}_n^\tau$ is the prediction of the $\tau$-rotated system, $\tau \in \mathrm{O}(3)^d$. The $\mathrm{O}(3)^d$ symmetry is restored as $n \to \infty$. As such, for large $n$, the system is approximately $\mathrm{O}(3)^d$-invariant. We randomly sample $\tau \in \mathrm{O}(3)^d$ and use the exponential map to construct a continuous interpolation $\tau_t \in \mathrm{O}(3)^d$ with $\tau_0 = \mathbf{Id}$ and $\tau_1 = \tau$. We train the network as in NN++ ($+\mathsf{LR}+\mathsf{L2}-\mathsf{C}$) using different $n$ and $\tau_t$ and average the predictions over 10 random initialization as an approximation of $\mathbb{E}\mathsf{VEC}_n^{\tau_t}(x)$. Not surprisingly, as $n$ increases and/or $t$ increases, (1) test performance decays monotonically (left panel in Fig.2), (2) the distance to $\mathbb{E}\mathsf{GAP}_n$ increases monotonically (middle panel) and (3) distance to $\mathsf{VEC}_\infty$ decrease monotonically (right panel). Clearly, the coordinate information from the data is utilized by smaller width $\mathsf{VEC}_n$.

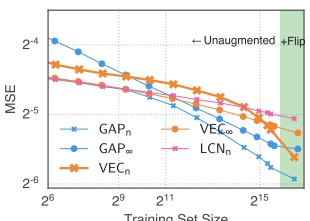

Figure 3: **Data Bends Learning Curve of** $\mathsf{VEC}_n$**.** We study the effect of training set size to the network's performance for various models. In the small dataset regime, the slope of the learning curve (in the log-log plot) of $\mathsf{VEC}_n$ is similar to that of $\mathsf{VEC}_\infty$ and $\mathsf{FCN}_n$. However, as the dataset gets larger, the slope increases significantly. This is hinted by Theorem 2.1.

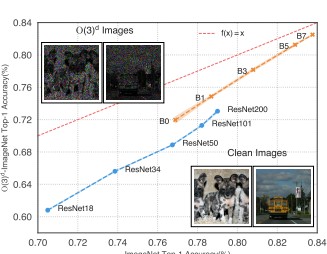 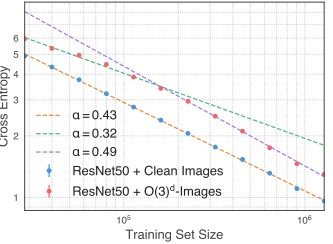 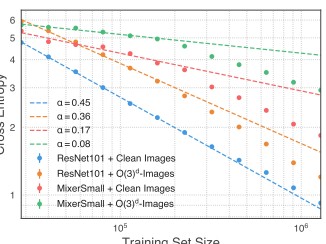

Figure 4: With coordinate of the input data rotated by $O(3)^d$, state of the art models learn as good as without rotation. middle/right: slopes of the learning curves increases due to more data. DIDE

**DIDE for $\mathsf{VEC}_n$.** To understand the role of data, we vary the training set size of Cifar10 from about $2^6$ to 50k (the whole un-augmented training set) and to 100k (adding left-right flip augmentation) and plot the learning curves in Fig.3. We observe dramatic speedup of learning for $\mathsf{VEC}_n$ in the larger data set regime, which isn't the case for $\mathsf{VEC}_\infty$ (kernel), $\mathsf{LCN}_n$, $\mathsf{GAP}_\infty$ and even for $\mathsf{GAP}_n$ after $m = 2^{12}$. We argue that this is due to the prior (the function space defined by the model) is too large (and not optimal) for the task and the coupled effect of more data together with inference procedures corrects the prior, as it is suggested by Theorem 2.1.

**DIDE for SOTA models.** In the middle and right panels of Fig.S2, we provide additional evidence in a larger scale setting. We generate learning curves of ImageNet using ResNet50 and MLP-Mixer, a very recent architecture that contains no convolution layers except the first layer, which is a convolution with filter size and stride equal to $(16, 16)$ (patches are disjoint). The symmetry group associated to ResNet is similar to that of $\mathsf{GAP}_n$ which is relatively small. However, the symmetry group induced by the first layer of the Mixer is $O(3 \times 16^2) \otimes \mathbf{I}_{14^2}$, where $3 \times 16^2$ is number of entries in the $(16, 16, 3)$ patch (RGB channels) and $14^2 = 224^2/16^2$ is the number of patches. Although the dimension of $O(3 \times 16^2) \otimes \mathbf{I}_{14^2}$ is quite large (about $(3 \times 16^2)^2/2$), it is still dramatically smaller than that of applying a fully-connected layer to the flatten images, which $O(3 \times 224^2)$ (about $(3 \times 224^2)^2/2$). In the middle panel of Fig.S2, we observe an almost perfect power-law scaling for the learning curve for the ResNet50 system with unrotated images. When the images are rotated by $O(3)^d$ ($d = 224^2$), the learning curve is relatively flat in the smaller data regime (green dashed line). However, the data set grows, it eventually catches up (purple dashed line) as that of the unrotated setting; see Sec.E for ResNet34/101. In the third panel, we see the learning curves are much flatter (red) for the Mixer and even more so for the rotated images (green). Again, these curves are bent towards that of ResNet50 with unrorated images as data increases, indicating the prior was being corrected.

Finally, in the left panel of Fig.S2, we compare the accuracy of state-of-the-art models trained on both unrotated and $O(3)^d$ rotated images. Surprisingly, the gap between the two are not large and becomes smaller for better performant models. For EfficientNet B7 [1], the top-1 accuracy of the rotated system is only 1.2% off from the unroated one.

## 4 Eigenecomposition of Neural Kernels

To get insights into the inductive biases, we eigendecompose the kernels using spherical harmonics. We assume the input space $\mathcal{X} = \{\xi = (\xi_0, \ldots, \xi_{p-1}) \in (\sqrt{d_0}\mathbb{S}^{(d_0-1)})^p\} \subseteq \mathbb{R}^{d_0 p}$, i.e.

---

[1] Still under training

the $p$-product of $(d_0 - 1)$-sphere with radius $\sqrt{d_0}$. We call $\xi_i \in \sqrt{d_0}\mathbb{S}^{(d_0-1)}$ a mini-patch and $(\xi_i, \xi_{i+1}, \ldots, \xi_{i+s-1}) \in (\sqrt{d_0}\mathbb{S}^{(d_0-1)})^s\}$ a patch for $i \in [p]$, where circular boundary condition is assumed. We consider the asymptotic limit when $d_0 = d^\alpha, p = d^{1-\alpha}$ and $d = pd_0 \to \infty$ and treat $0 < \alpha < 1$ and $s$ as fixed constant. The input space $\mathcal{X}$ is associated with the product measure $\mu \equiv \sigma_{d_0}^p$, where $\sigma_{d_0}$ is the normalized uniform measure on $\sqrt{d_0}\mathbb{S}^{(d_0-1)}$. The kernels associated to the one-hidden layer infinite networks (either NNGP or NTK) has the following general forms

$$k\left(\frac{1}{p}\sum_{i\in[p]}\xi_i^T\eta_i/d_0\right) \quad \frac{1}{p}\sum_{i\in[p]}k\left(\frac{1}{s}\sum_{b\in[s]}\xi_{i+b}^T\eta_{i+b}/d_0\right) \quad \frac{1}{p^2}\sum_{i,j\in[p]}k\left(\frac{1}{s}\sum_{b\in[s]}\xi_{i+b}^T\eta_{j+b}/d_0\right),$$
(12)

although that exact form of the (positive definite) kernel function $k : \mathbb{R} \to \mathbb{R}$ depends on the kernel types (NNGP vs NTK), activations, hyperparameters and etc. We assume the kernel is sufficiently smooth in $(-1, 1)$ and the Tayor expansion of $k^{(r)}$ converges uniformly in $[-1, 1]$ for sufficiently many $r \in \mathbb{N}$. We use the notation that $A \sim B$ if there are positive constants $c$ and $C$ such that $cA \leq B \leq CA$ for $d$ sufficiently large. We use $\mathcal{K}$ to represent any kernels above and consider it as a Hilbert–Schmidt operator on $L^2(\mathcal{X}, \mu)$

$$\mathcal{K}f(\xi) = \int_\mathcal{X} \mathcal{K}(\xi, \eta)f(\eta)d\mu, \quad f \in L^2(\mathcal{X}, \mu),$$
(13)

which is well-defined since $\mu$ is a probability measure and $k$ is bounded. Let $\vec{r} = (r_0, \ldots, r_{p-1}) \in \mathbb{N}^p$, $\tau$ the shifting operator $\tau\vec{r} = (r_{p-1}, r_0, \ldots, r_{p-2})$. The $s$-banded subset of $\mathbb{N}^p$ is defined to be

$$B(\mathbb{N}^p, s) = \{\vec{r} \in \mathbb{N}^p : \text{dist}(\text{argmax}_j r_j \neq 0, \text{argmin}_j r_j \neq 0) \leq s - 1\}$$
(14)

which is a quantifier used to restrict the support of a function on a patch. Here $\text{dist}(i, j) = \min\{|i - j|, p - |i - j|\}$, a distance defined on the cyclic group $[p] = \mathbb{Z}/p\mathbb{Z}$. The quotient space $B(\mathbb{N}^p, s)/\tau$ denotes a subset of $B(\mathbb{N}^p, s)$ by identifying $\vec{v} = \vec{v}'$ as the same element if $\vec{v} = \tau^a\vec{r}'$ for some $a \in [p]$. Finally, $Y_{r_j,l_j}(\xi_j)$ is used to denote the $l_j$-th spherical harmonic of degree $r_j$ in the unit sphere $\mathbb{S}^{(d_0-1)}$ and has unit norm under the normalized measure on $\mathbb{S}^{(d_0-1)}$. As such $Y_{r_j,l_j}(\xi_j/\sqrt{d_0}) \in L^2(\sqrt{d_0}\mathbb{S}^{(d_0-1)}, \sigma_{d_0})$ has unit norm. Recall that the total number of spherical harmonic of degree $r_j$ in $\mathbb{S}^{(d_0-1)}$ is $N(d_0, r_j) = (2r_j + d_0 - 2)\binom{d_0+r_j-3}{r_j-1} \sim d_0^{r_j}/r_j!$ as $d_0 \to \infty$. We use $N(d_0, \vec{r}) = \prod_{j\in[p]} N(d_0, r_j)$ and $[N(d_0, \vec{r})] = \prod_{j\in[p]}[N(d_0, r_j)]$, resp. Let

$$\vec{Y}_{\vec{r},\vec{l}}(\xi) = \prod_{j\in[p]} Y_{r_j,l_j}(\xi_j)$$
(15)

The following theorem shows that locality ($\mathsf{VEC}_\infty$) dramatically reduces both the dimensions of Eigendecomposition $r \geq 1$ eigenspaces and the spectral gap between them. In addition, pooling (i.e. translation symmetry of $\mathsf{GAP}_n$) reduces their dimensions by a factor of $p$. See Sec.E for the implication of this theorem to learning.

**Theorem 4.1.** *[Sec.D] We have the following eigendecomposition for the integral operator $\mathcal{K}$*

$$\mathsf{H} = \bigcup_{r\in\mathbb{N}} \mathsf{H}^{(r)} = \bigcup_{r\in\mathbb{N}} \bigcup_{\vec{r}\in Q(\mathcal{K},r)} \mathsf{H}^{(\vec{r})},$$
(16)

*where $Q(\mathcal{K}, r)$ is a quantifier defined below. If $r = 0$, $\mathsf{H}^{(0)}$ is the space of constant functions and the eigenvalue is $\sim k(0)$. For $r \geq 1$,*

    *1. if $\mathcal{K} = \mathcal{K}_{\mathsf{FCN}}$, then $Q(\mathcal{K}, r) = \{\vec{r} \in \mathbb{N}^p : |\vec{r}| = r\}$ and the unit eigenfunctions are*

$$\begin{cases} \mathsf{H}^{(\vec{r})} = \text{span}\left\{Y_{\vec{r},\vec{l}}\right\}_{\vec{l}\in[B(d_0,\vec{r})]} \\ \dim(\mathsf{H}^{(r)}) \sim d^r \quad and \quad \lambda(\mathsf{H}^{(\vec{r})}) \sim d^{-r}\delta(k^{(r)}(0)) \end{cases}$$
(17)

    *2. if $\mathcal{K} = \mathcal{K}_{\mathsf{VEC}}$, $Q(\mathcal{K}, r) = \{\vec{r} \in B(\mathbb{N}^p, s) : |\vec{r}| = r\}$ the unit eigenfunctions are*

$$\begin{cases} \mathsf{H}_{\mathsf{VEC}}^{(\vec{r})} = \text{span}\left\{Y_{\vec{r},\vec{l}}\right\}_{\vec{l}\in[B(d_0,\vec{r})]} \\ \dim(\mathsf{H}_{\mathsf{VEC}}^{(r)}) \sim p(sd_0)^r = s^r d^{1-\alpha+r\alpha} \quad and \quad \lambda(\mathsf{H}_{\mathsf{VEC}}^{(\vec{r})}) \sim p^{-1}(sd_0)^{-r}\delta(k^{(r)}(0)) \end{cases}$$
(18)

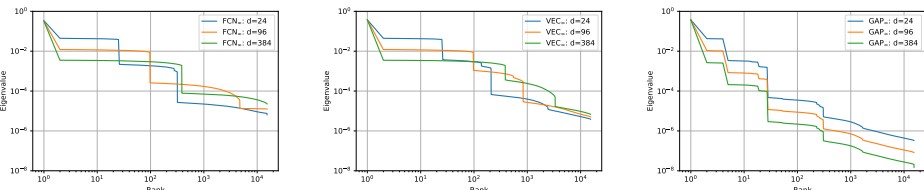

Figure 5: **Eigenvalue Decay of Relu NTK of** $\mathsf{FCN}_\infty$, $\mathsf{VEC}_\infty$ **and** $\mathsf{GAP}_\infty$. $d_0 = s = 3$. The eigenvalues of $\mathsf{GAP}_\infty$ decays *faster* because with $m = 15k$ many samples, higher order eigenspace can be covered by $\mathsf{GAP}_\infty$ but not $\mathsf{FCN}_\infty/\mathsf{VEC}_\infty$ due to Theorem 4.1.

3. *and finally, if $\mathcal{K} = \mathcal{K}_{\mathsf{GAP}}$, then $Q(\mathcal{K}, r) = \{\vec{r} \in B(\mathbb{N}^p, s)/\tau : |\vec{r}| = r\}$, the unit eigenfunctions are*

$$\begin{cases} \mathsf{H}_{\mathsf{GAP}}^{(\vec{r})} = \mathrm{span}\left\{ \frac{1}{\sqrt{p}} \sum_{\tau \in [p]} Y_{\vec{r}, \vec{l}}(\tau \xi) \right\}_{\vec{l} \in [B(d_0, \vec{r})]} \\ \dim(\mathsf{H}_{\mathsf{GAP}}^{(r)}) \sim (sd_0)^r = s^r d^{r\alpha} \quad and \quad \lambda(\mathsf{H}_{\mathsf{GAP}}^{(\vec{r})}) \sim p^{-1}(sd_0)^{-r} \delta(k^{(r)}(0)) \end{cases} \tag{19}$$

# 5  Related Work

The study of infinite networks dates back to seminal work by Neal [8] who showed the convergence of single hidden-layer networks to Gaussian Processes (GPs). Recently, there has been renewed interest in studying random, infinite, networks starting with concurrent work on "conjugate kernels" [10, 35] and "mean-field theory" [9, 36], taking a statistical learning and statistical physics view of points, resp. Since then this analysis has been extended to include a wide range for architectures [20, 21, 37, 29, 26, 38]. The inducing kernel is often referred to as the Neural Network Gaussian Process (NNGP) kernel. The neural tangent kernel (NTK), first introduced in Jacot et al. [22], along with followup work [12, 39] showed that the distribution of functions induced by gradient descent for infinite-width networks is a Gaussian Process with NTK as the kernel.

The study of implicit bias (regularization) of gradient descent has received considerable interests. The work [15, 40–43] demonstrate the convergence of SGD to the maximal margin solution for logistic-type of losses during late time training. [44–50] study the early-time SGD dynamics, spectral biases of neural networks. These results aim to explain the order of learning of neural networks: functions of less complexity are usually learned before more complex functions.

[27] is the first to show that the prediction functions obtained from training FCN depend, in addition on the labels, only on the covariance of the input data. This implies our result regarding the $\mathrm{O}(3d)$ invariance of FCN. By utilizing this symmetry, recent work [51] constructs a particular task where the label function is a second order polynomial of the inputs and show that orthogonal invariance algorithm requires sample size of order $d^2$ while there is a convnet requires only $O(1)$ samples. Their convnet essentially corresponds to the $d_0 = s = 1$ and $r = 2$ case of Theorem 4.1, in which the dimension of this eigenspace (and indeed of all $r$-eigenspace by treating $r$ as a finite constant as $d \to \infty$) of $\mathsf{GAP}_\infty$ is $O(1)$ while the dimension of the 2-eigenspace of $\mathsf{FCN}_\infty$ is of order $d^2$.

# 6  Conclusion

In this paper, we consider machine learning methods as an integrated system of data, models and inference algorithms and study the basic symmetries of various machine learning systems. We surface the importance of locality in modern machine learning systems through large scale empirical study and through an eigendecomposition of one-layer infinite networks. However, we haven't addressed the two import questions (1) theoretical characterization of the effect of composing locality and (2) the mathematical understanding of DIDE and how the prior is corrected by the coupled effect of data and gradient descent. We leave them to future work.

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
