## 4 Eigenecomposition of Neural Kernels

To gain insight into the inductive biases of various architectures, we eigendecompose the kernels using spherical harmonics. We assume the input space $\mathcal{X} = \{\xi = (\xi_0, \ldots, \xi_{p-1}) \in (\sqrt{d_0}\mathbb{S}^{(d_0-1)})^p\} \subseteq$

---

[1]Still under training

$\mathbb{R}^{d_0 p}$, i.e. the $p$-product of $(d_0 - 1)$-sphere with radius $\sqrt{d_0}$. We call $\xi_i \in \sqrt{d_0}\mathbb{S}^{(d_0-1)}$ a mini-patch and $(\xi_i, \xi_{i+1}, \ldots, \xi_{i+s-1}) \in (\sqrt{d_0}\mathbb{S}^{(d_0-1)})^s\}$ a patch for $i \in [p]$, where circular boundary condition is assumed. We consider the asymptotic limit when $d_0 = d^\alpha, p = d^{1-\alpha}$ and $d = pd_0 \to \infty$ and treat $0 < \alpha < 1$ and $s$ as fixed constant. The input space $\mathcal{X}$ is associated with the product measure $\mu \equiv \sigma_{d_0}^p$, where $\sigma_{d_0}$ is the normalized uniform measure on $\sqrt{d_0}\mathbb{S}^{(d_0-1)}$. The kernels associated to the one-hidden layer infinite networks (NNGP and NTK) have the following general forms

$$
k\left(\frac{1}{p}\sum_{i\in[p]}\xi_i^T \eta_i/d_0\right), \quad \frac{1}{p}\sum_{i\in[p]} k\left(\frac{1}{s}\sum_{b\in[s]}\xi_{i+b}^T\eta_{i+b}/d_0\right), \quad \frac{1}{p^2}\sum_{i,j\in[p]} k\left(\frac{1}{s}\sum_{b\in[s]}\xi_{i+b}^T\eta_{j+b}/d_0\right),
$$
(12)

for $\mathcal{K}_{\mathsf{FCN}}, \mathcal{K}_{\mathsf{VEC}}$ and $\mathcal{K}_{\mathsf{GAP}}$, resp. Note that the

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

# Supplementary Material

## A  Glossary

We use the following abbreviations in this work:

- +L2: Adding L2 regularization.
- +LR: Using a large learning rate.
- +DA: Applying MixUp data augmentation.
- +C: Centering the outputs of the network.
- −C: Remove centering.
- $\mathsf{FCN}_n$:Fully-connected networks with width $n$.
- $\mathsf{FCN}_\infty$: Infinite width $\mathsf{FCN}_n$.
- $\mathsf{VEC}_n$: Convnet with width $n$ and a flattening readout layer.
- $\mathsf{VEC}_\infty$:Infinite width $\mathsf{VEC}_n$.
- $\mathsf{LCN}_n$: Locally-connected network with width $n$.
- $\mathsf{LCN}_\infty$: Infinite width $\mathsf{LCN}_n$, which is the samme as $\mathsf{VEC}_\infty$.
- $\mathsf{GAP}_n$: Convnet with width $n$ and a global average readout layer.
- $\mathsf{GAP}_\infty$:Infinite width $\mathsf{GAP}_n$.
- $\mathsf{LAP}_n^k$: Similar to $\mathsf{GAP}_n$, except the readout layer is a $(k, k)$ average pooling.
- $\mathsf{LAP}_\infty^k$: Infinite width $\mathsf{LAP}_n^k$.

## B  Proof of Theorem 2.1

We use $\mathsf{FCN}_n$ to denote the class of functions that can be expressed by $L$-hidden layer fully-connected networks whose widths are equal to $n$. Similar notation applies to other architectures.

**Corollary 1.** *We have the following*

$$\mathsf{GAP}_n \subseteq \mathsf{VEC}_n \subseteq \mathsf{LCN}_n \subseteq \mathsf{VEC}_{dn}, \quad \mathsf{LCN}_n \subseteq \mathsf{FCN}_{dn} \tag{S1}$$

*Proof.* We only need to prove $\mathsf{LCN}_n \subseteq \mathsf{VEC}_{dn}$ because the others are obvious. Let $\mathsf{LCN}_n(x)^l_{\alpha,i}$ denote the post-activation at layer $l$, spatial location $\alpha$ and channel index $i$ of a $\mathsf{LCN}_n$ with input $x$ and $\mathsf{VEC}_n(x)^l_{\alpha,i}$ is defined similarly. It suffices to prove that for any LCN with width $n$ there is a VEC with width $dn$ such that for any $l \geq 1$ (i.e. not the input layer)

$$\mathsf{VEC}_{dn}(x)^l_{\alpha,\alpha n+i} = \mathsf{LCN}_n(x)^l_{\alpha,i} \tag{S2}$$

since we could choose the readout weights of $\mathsf{VEC}_{dn}$ at locations $(\alpha, \alpha n + i)$ to match the one of $\mathsf{LCN}_n$ at locaton $(\alpha, i)$ and zero out the remaining entries. We prove this by induction and assume it holds for $l$ (the base case $l = 1$ is obvious). Then the $\mathsf{LCN}_n$ and $\mathsf{VEC}_n$ at layer $l + 1$ can be written as

$$\mathsf{LCN}_n(x)^{l+1}_{\alpha,j} = \phi\left( \frac{1}{\sqrt{n(2k+1)}} \sum_{i\in[n],\beta\in[-k,k]} \mathsf{LCN}_n(x)^l_{\alpha+\beta,i}\omega^{l+1}_{\beta,ij}(\alpha) \right)$$

and

$$\mathsf{VEC}_{dn}(x)^{l+1}_{\alpha,j} = \phi\left( \frac{1}{\sqrt{dn(2k+1)}} \sum_{i\in[dn],\beta\in[-k,k]} \mathsf{VEC}_{dn}(x)^l_{\alpha+\beta,i}\tilde{\omega}^{l+1}_{\beta,ij} \right)$$

One can show that Equation S2 holds for $(l + 1)$ by choosing the parameters of $\mathsf{VEC}_{dn}$ as follows

$$\tilde{\omega}^{l+1}_{\beta,ij} = \sqrt{d}\omega^{l+1}_{\beta,i-(\alpha+\beta)n,j-\alpha n} \quad \text{if} \quad \alpha n \leq j < \alpha(n+1) \quad \text{and} \quad (\alpha+\beta)n \leq i < (\alpha+\beta)(n+1)$$

and 0 otherwise.

$\square$

## C Proof of Symmetries

*Proof.* For simplicity, we present the proof for full-batch training. The proof applies to mini-batch training as long as order of the mini-batch is fixed. Let $\tau$ be a rotation in $O(3d)$ or $O(3)^d$ or $O(3) \otimes \mathbf{I}_d$, depending on the architectures ($\mathsf{FCN}_n, \mathsf{LCN}_n, \mathsf{VEC}_n, \mathsf{GAP}_n$) and the tuple $\theta$ and $\gamma$ denote the parameters of the first and remaining layers of the network, resp. Let $h(\tau x, \theta) = \langle \tau x, \theta \rangle$ denote the pre-activations of the first-hidden layer in the rotated coordinate. Here $\langle \cdot, \cdot \rangle$ is the bilinear map (a dense layer or a convolutional layer with or without weight-sharing, etc.), not the inner product. The loss with $L^2$-regularization is

$$R_\lambda(\theta, \gamma) = L(h(\tau\mathcal{X}, \theta), \gamma) + \frac{1}{2}\lambda(\|\theta\|_2^2 + \|\gamma\|_2^2) \tag{S3}$$

where $L(h(\tau\mathcal{X}, \theta), \gamma)$ is the raw loss of the network. For each random instantiation $\theta = \theta_0$ with $\theta_0$ drawn from standard Gaussian iid, we instantiate a coupled network from the un-rotated coordinates but with a different instantiation in the first layer $\theta^\tau = \tau^*\theta_0$ and keep the remaining layers unchanged, i.e. $\gamma^\tau = \gamma_0$. Here $\tau^*$ is the adjoint of $\tau$ and note that $\tau^*\theta_0$ and $\theta_0$ have the same distribution by the Gaussian initialization of $\theta_0$ and the definition of $\tau$. The regularized loss associated to this instantiation is

$$R_\lambda(\theta^\tau, \gamma^\tau) = L(h(\mathcal{X}, \theta^\tau), \gamma^\tau) + \frac{1}{2}\lambda(\|\theta^\tau\|_2^2 + \|\gamma^\tau\|_2^2) \tag{S4}$$

It suffices to prove that for each instantiation $\theta = \theta_0$ drawn from Gaussian, the following holds for all gradient steps $t$

$$(\theta_t^\tau, \gamma_t^\tau) = (\tau^*\theta_t, \gamma_t). \tag{S5}$$

We prove this by induction on $t$ and $t = 0$ is true by definition. Assume it holds when $t = t$. Now the update in $\gamma$ and $\gamma^\tau$ with learning rate $\eta$ are

$$\gamma_{t+1} = \gamma_t - \eta\left(\left.\frac{\partial L}{\partial \gamma}\right|_{(h(\tau\mathcal{X}, \theta_t), \gamma_t)}\right)^T - \eta\lambda\gamma_t \tag{S6}$$

$$\gamma_{t+1}^\tau = \gamma_t^\tau - \eta\left(\left.\frac{\partial L}{\partial \gamma}\right|_{(h(\mathcal{X}, \theta_t^\tau), \gamma_t^\tau)}\right)^T - \eta\lambda\gamma_t^\tau \tag{S7}$$

It is clear $\gamma_{t+1} = \gamma_{t+1}^\tau$ by induction since $h(\tau\mathcal{X}, \theta_t) = h(\mathcal{X}, \theta_t^\tau)$. Similarly,

$$\theta_{t+1} = \theta_t - \eta\left(\left.\frac{\partial L}{\partial h}\frac{\partial h}{\partial \theta}\right|_{(\tau\mathcal{X}, \theta_t))}\right)^T - \lambda\theta_t \tag{S8}$$

$$\theta_{t+1}^\tau = \theta_t^\tau - \eta\left(\left.\frac{\partial L}{\partial h}\frac{\partial h}{\partial \theta^\tau}\right|_{(\mathcal{X}, \theta_t^\tau)}\right)^T - \lambda\theta^\tau \tag{S9}$$

Note that by the chain rule and induction assumption

$$\left.\frac{\partial h}{\partial \theta^\tau}\right|_{(\mathcal{X}, \theta_t^\tau)} = \left.\frac{\partial h}{\partial \theta}\right|_{(\mathcal{X}, \theta_t^\tau)}\frac{\partial \theta^\tau}{\partial \theta} = \left.\frac{\partial h}{\partial \theta}\right|_{(\mathcal{X}, \theta_t^\tau)}\tau \tag{S10}$$

This implies $\theta_{t+1}^\tau = \tau^*\theta_{t+1}$.

$\square$

**Remark S1.** It is not difficult to see the apply proof apply to Non-Gaussian i.i.d. initialization (e.g. uniform distribution) and/or adding $L^p$-regularization when the rotation groups are replaced by the corresponding permutation groups. Empirically, we observe that replacing the first layer Gaussian initialization by uniform distribution does not change the performance of the network much. See Fig.S2.

**Remark S2.** The proof works for all parameterization methods, including NTK-parameterization[11], standard parameterization [52], mean-field parameterization[53] and ABC-parameterization [54]

 # D Eigendecomposition of Infinite Networks

To gain insight into the inductive biases of various architectures, we eigendecompose the kernels using spherical harmonics. We assume the input space $\mathcal{X} = \{\xi = (\xi_0, \ldots, \xi_{p-1}) \in (\sqrt{d_0}\mathbb{S}^{(d_0-1)})^p\} \subseteq \mathbb{R}^{d_0 p}$, i.e. the $p$-product of $(d_0 - 1)$-sphere with radius $\sqrt{d_0}$. We call $\xi_i \in \sqrt{d_0}\mathbb{S}^{(d_0-1)}$ a mini-patch and $(\xi_i, \xi_{i+1}, \ldots, \xi_{i+s-1}) \in (\sqrt{d_0}\mathbb{S}^{(d_0-1)})^s\}$ a patch for $i \in [p]$, where circular boundary condition is assumed. We consider the asymptotic limit when $d_0 = d^\alpha$, $p = d^{1-\alpha}$ and $d = pd_0 \to \infty$ and treat $0 < \alpha < 1$ and $s$ as fixed constant.

The input space $\mathcal{X}$ is associated with the product measure $\mu \equiv \sigma_{d_0}^p$, where $\sigma_{d_0}$ is the normalized uniform measure on $\sqrt{d_0}\mathbb{S}^{(d_0-1)}$. The kernels associated to the one-hidden layer infinite networks (NNGP and NTK) have the following general forms

$$\mathcal{K}_{\mathsf{FCN}}(\xi, \eta) = k\left(\frac{1}{p} \sum_{i \in [p]} \xi_i^T \eta_i / d_0\right), \tag{S11}$$

$$\mathcal{K}_{\mathsf{VEC}}(\xi, \eta) = \frac{1}{p} \sum_{i \in [p]} k\left(\frac{1}{s} \sum_{b \in [s]} \xi_{i+b}^T \eta_{i+b} / d_0\right), \tag{S12}$$

$$\mathcal{K}_{\mathsf{GAP}}(\xi, \eta) = \frac{1}{p^2} \sum_{i,j \in [p]} k\left(\frac{1}{s} \sum_{b \in [s]} \xi_{i+b}^T \eta_{j+b} / d_0\right), \tag{S13}$$

for $\mathsf{FCN}_\infty$, $\mathsf{VEC}_\infty$ and $\mathsf{GAP}_\infty$ resp. Note that the exact form of the (positive definite) kernel function $k : \mathbb{R} \to \mathbb{R}$ depends on the type of the kernels (NNGP vs NTK), activations, hyperparameters and etc. We assume the kernel is sufficiently smooth in $(-1, 1)$ and the Tayor expansion of $k^{(r)}$ converges uniformly in $[-1, 1]$ for sufficiently many $r \in \mathbb{N}$. We use the notation that $A \sim B$ if there are positive constants $c$ and $C$ such that $cA \leq B \leq CA$ for $d$ sufficiently large. We use $\mathcal{K}$ to represent any kernels above and consider it as a Hilbert–Schmidt operator on $L^2(\mathcal{X}, \mu)$

$$\mathcal{K}f(\xi) = \int_{\mathcal{X}} \mathcal{K}(\xi, \eta) f(\eta) d\mu, \quad f \in L^2(\mathcal{X}, \mu), \tag{S14}$$

which is well-defined since $\mu$ is a probability measure and $k$ is bounded.

## D.1 Legendre Polynomials, Spherical Harmonics and their Tensor Products.

Our notation follows closely from [55], an excellent introduction to spherical harmonics.

**Legendre Polynomials.** Let $\omega_{d_0}$ be the measure defined on the interval $I = [-1, 1]$

$$\omega_{d_0}(t) = (1 - t^2)^{(d_0-3)/2} \tag{S15}$$

The Legendre polynomials[2] $\{P_r(t) : r \in \mathbb{N}\}$ is an orthogonal basis for the Hilbert space $L^2(I, \omega_{d_0})$, i.e.

$$\int_I P_r(t) P_{r'}(t) \omega_{d_0}(t) dt = 0 \quad \text{if} \quad r \neq r' \tag{S16}$$

Here $P_r(t)$ is a degree $r$ polynomials with $P_r(1) = 1$ and satisfies the Rodrigues formula

**Lemma 1** (Rodrigues Formula. Proposition 4.19 [55])**.**

$$P_r(t) = c_r \omega_{d_0}^{-1} \left(\frac{d}{dt}\right)^r (1 - t^2)^{r+(d_0-3)/2}, \tag{S17}$$

*where*

$$c_r = \frac{(-1)^r}{2^r (r + (d_0 - 3)/2)_r} \tag{S18}$$

---

[2]More accurate, this should be called Gegenbauer Polynomials. However, we decide to stick to the terminology in [55]

In the above lemma, $(x)_l$ denotes the falling factorial

$$(x)_l \equiv x(x-1)\cdots(x-l+1) \tag{S19}$$

$$(x)_0 \equiv 1 \tag{S20}$$

**Spherical Harmonics.** Let $d\mathbb{S}_{d_0-1}$ define the (un-normalized) uniform measure on the unit sphere $\mathbb{S}_{d_0-1}$. Then

$$|\mathbb{S}_{d_0-1}| \equiv \int_{\mathbb{S}_{d_0-1}} d\mathbb{S}_{d_0-1} = \frac{2\pi^{d_0/2}}{\Gamma(\frac{d_0}{2})} \tag{S21}$$

The normalized measure on this sphere is defined to be

$$d\sigma_{d_0} = \frac{1}{|\mathbb{S}_{d_0-1}|} d\mathbb{S}_{d_0-1} \quad \text{and} \quad \int_{\mathbb{S}_{d_0-1}} d\sigma_{d_0} = 1 \tag{S22}$$

The spherical harmonics $\{Y_{r,l}\}_{r,l}$ in $\mathbb{R}^{d_0}$ are homogeneous harmonic polynomials that form an orthonormal basis in $L^2(\mathbb{S}_{d_0-1}, \sigma_{d_0})$

$$\int_{\xi\in\mathbb{S}_{d_0-1}} Y_{r,l}(\xi)Y_{r',l'}(\xi)d\sigma_{d_0} = \delta_{(r,l)=(r',l')}. \tag{S23}$$

Here $Y_{r,l}$ denotes the $l$-th spherical harmonic whose degree is $r$, where $r \in \mathbb{N}$, $l \in [N(d_0, r)]$ and

$$N(d_0, r) = \frac{2r + d_0 - 2}{r}\binom{d_0 + r - 3}{r - 1} \sim d_0^r/r! \quad \text{as} \quad d_0 \to \infty. \tag{S24}$$

The Legendre polynomials and spherical harmonics are related through the addition theorem.

**Lemma 2** (Addition Theorem. Theorem 4.11 [55])**.**

$$P_r(\xi^T\eta) = \frac{1}{N(d_0, r)} \sum_{l\in[N(d_0,r)]} Y_{r,l}(\xi)Y_{r,l}(\eta), \quad \xi, \eta \in \mathbb{S}_{d_0-1}. \tag{S25}$$

**Tensor Products.** Let $p \in \mathbb{N}$, $\vec{r} \in \mathbb{N}^p$, $I^p = [-1,1]^p$ and $\omega_{d_0}^p$ be the product measure on $I^p$. Then the (product of) Legendre polynomials

$$P_{\vec{r}}(\vec{t}) = \prod_{j\in[p]} P_{r_j}(t_j), \quad \vec{t} = (t_1, \ldots, t_p) \in I^p \tag{S26}$$

form an orthogonal basis for the Hilbert space $L^2(I^p, \omega_{d_0}^p) = (L^2(I, \omega_{d_0}))^{\otimes p}$. Similarly, the product of spherical harmonics

$$Y_{\vec{r},\vec{l}} = \prod_{j\in[p]} Y_{r_j,l_j}, \quad \vec{l} = (l_1, \ldots, l_p) \in [N(d_0, \vec{r})] \equiv \prod_{j\in p}[N(d_0, r_j)] \tag{S27}$$

form an orthonormal basis for the product space

$$L^2(\mathbb{S}_{d_0-1}^p, \sigma_{d_0}^p) = (L^2(\mathbb{S}_{d_0-1}, \sigma_{d_0}))^{\otimes p}. \tag{S28}$$

Elements in the set $\{Y_{\vec{r},\vec{l}}\}_{\vec{l}\in[B(d_0,\vec{r})]}$ are called degree (order) $\vec{r}$ spherical harmonics in $L^2(\mathbb{S}_{d_0-1}^p, \sigma_{d_0}^p)$ and also degree $r$ spherical harmonics if $|\vec{r}| = r \in \mathbb{N}$.

## D.2 "Fourier" Decomposition.

Let $K(\vec{t}) \in L^2(\mathbb{S}_{d_0-1}^p, \sigma_{d_0}^p)$. Then we have the following "Fourier decomposition" (the convergence is in $L^2$),

$$K(\vec{t}) = \sum_{\vec{r}\in\mathbb{N}^p} \hat{K}(\vec{r})P_{\vec{r}}(\vec{t}) \tag{S29}$$

where the "Fourier coefficients" are

$$\hat{K}(\vec{r}) = \langle K, P_{\vec{r}}\rangle_{L^2(I^p, \omega_{d_0}^p)} / \langle P_{\vec{r}}, P_{\vec{r}}\rangle_{L^2(I^p, \omega_{d_0}^p)}. \tag{S30}$$

634 Applying Lemma 2, we have the harmonic decomposition (the convergence is in $L^2$)

$$K(\xi^T \eta) = \sum_{\vec{r} \in \mathbb{N}^p} \hat{K}(\vec{r}) N(d_0, \vec{r})^{-1} \sum_{\vec{l} \in [B(d_0, \vec{r})]} Y_{\vec{r}, \vec{l}}(\xi) Y_{\vec{r}, \vec{l}}(\eta), \quad \xi, \eta \in \mathbb{S}_{d_0 - 1}^p \qquad (S31)$$

635 Clearly, as an integral operator

$$\int_{\mathbb{S}_{d_0-1}^p} K(\xi^T \eta) Y_{\vec{r}, \vec{l}}(\eta) d\sigma_{d_0}^p = \hat{K}(\vec{r}) N(d_0, \vec{r})^{-1} Y_{\vec{r}, \vec{l}}(\xi). \qquad (S32)$$

636 **Theorem D.1.** *Let $K(\vec{t}) \in L^2(\mathbb{S}_{d_0-1}^p, \sigma_{d_0}^p)$. Then.*

$$K(\xi^T \eta) = \sum_{\vec{r} \in \mathbb{N}^p} \hat{K}(\vec{r}) N(d_0, \vec{r})^{-1} \sum_{\vec{l} \in [B(d_0, \vec{r})]} Y_{\vec{r}, \vec{l}}(\xi) Y_{\vec{r}, \vec{l}}(\eta), \quad \xi, \eta \in \mathbb{S}_{d_0 - 1}^p \qquad (S33)$$

637 *If in addition $\|K\|_{C^{|\vec{r}|+1}(I^p)} < \infty$, then*

$$\hat{K}(\vec{r}) = \vec{r}!^{-1} \left( K^{(\vec{r})}(0) + \mathcal{O}(\|K\|_{C^{|\vec{r}|+1}(I^p)} p d_0^{-\frac{1}{2}}) \right). \qquad (S34)$$

638 Therefore, the eigenvalues of $K(\xi^T \eta)$ are $\hat{K}(\vec{r}) N(d_0, \vec{r})^{-1}$, with eigenspace spanned by the (unit)
639 eigenvectors $\{Y_{\vec{r}, \vec{l}}\}_{\vec{l} \in [N(d_0, \vec{r})]}$ whose dimension is $N(d_0, \vec{r})$, resp.

## D.3  Eigendecomposing the Infinite Networks

641 To handle the patch, we introduce the $s$-banded subset of $\mathbb{N}^p$. For $i, j \in [p]$, define the a distance in
642 the cyclic group $[p] = \mathbb{Z}/p\mathbb{Z}$ to be

$$\mathrm{dist}(i, j) = \min\{|i - j|, p - |i - j|\},$$

643 and the diameter of $\vec{r} \in \mathbb{N}^p$ to be

$$\mathrm{diam}(\vec{r}) = \mathrm{dist}(\mathrm{argmax}_j r_j \neq 0, \mathrm{argmin}_j r_j \neq 0) \qquad (S35)$$

644 The $s$-banded subset of $\mathbb{N}^p$ is the collection of points whose diameter is less than $s$, i.e.,

$$B(\mathbb{N}^p, s) = \{\vec{r} \in \mathbb{N}^p : \mathrm{diam}(r) \leq s - 1\} \qquad (S36)$$

645 This implies $Y_{\vec{r}, \vec{l}}$ is a function defined on a patch if and only if $\vec{r} \in B(\mathbb{N}^p, s)$.

646 Let $\tau$ be shifting operator $\tau \vec{r} = (r_{p-1}, r_0, \ldots, r_{p-2})$, where $\vec{r} = (r_0, \ldots, r_{p-1}) \in \mathbb{N}^p$. The quotient
647 space $B(\mathbb{N}^p, s)/\tau$ denotes a subset of $B(\mathbb{N}^p, s)$ by identifying $\vec{v} = \vec{v}'$ as the same element if $\vec{v} = \tau^a \vec{r}'$
648 for some $a \in [p]$.

649 In deep learning, it is more convenient to work on the non-unit sphere $\sqrt{d_0} \mathbb{S}_{d_0-1}$. We still use $\sigma_{d_0}$ to
650 denote the normalized (probability) measure on $\sqrt{d_0} \mathbb{S}_{d_0-1}$. The spherical harmonics with unit norms
651 are

$$Y_{r_j, l_j} \left( \frac{\xi_j}{\sqrt{d_0}} \right) \in L^2 \left( \sqrt{d_0} \mathbb{S}^{(d_0-1)}, \sigma_{d_0} \right) \qquad (S37)$$

$$Y_{\vec{r}, \vec{l}} \left( \frac{\xi}{\sqrt{d_0}} \right) \in L^2 \left( \left( \sqrt{d_0} \mathbb{S}^{(d_0-1)} \right)^p, \sigma_{d_0}^p \right) \qquad (S38)$$

652 The following theorem characterize the inductive biases induced by locality and symmetry (i.e.
653 shifting invariant) for infinite networks. It shows that locality ($\mathsf{VEC}_\infty$) dramatically reduces both
654 the dimensions of $r \geq 1$ eigenspaces and the spectral gap among them. In addition, pooling (i.e.
655 resulting shifting invariant for $\mathsf{GAP}_n$) reduces their dimensions by an additional factor of $p$. See
656 Sec.E for the implication of this theorem to learning.

657 **Theorem D.2.** *We have the following eigendecomposition for the integral operator $\mathcal{K}$*

$$\mathsf{H} = \bigcup_{r \in \mathbb{N}} \mathsf{H}^{(r)} = \bigcup_{r \in \mathbb{N}} \bigcup_{\vec{r} \in Q(\mathcal{K}, r)} \mathsf{H}^{(\vec{r})}, \qquad (S39)$$

658 *where $Q(\mathcal{K}, r)$ is a quantifier defined below. If $r = 0$, $\mathsf{H}^{(0)}$ is the space of constant functions and the*
659 *eigenvalue is $\sim k(0)$. For $r \geq 1$, we have the following.*

660 **(1)Base Case:** $\mathcal{K} = \mathcal{K}_{\text{FCN}}$. $Q(\mathcal{K}, r) = \{\vec{r} \in \mathbb{N}^p : |\vec{r}| = r\}$ *and the unit eigenfunctions are*

$$
\begin{cases}
\mathsf{H}^{(\vec{r})} = \text{span} \left\{ Y_{\vec{r},\vec{l}}(\frac{\cdot}{\sqrt{d_0}}) \right\}_{\vec{l} \in [B(d_0, \vec{r})]} \\
\dim(\mathsf{H}^{(r)}) \sim d^r \quad and \quad \lambda(\mathsf{H}^{(\vec{r})}) \sim d^{-r} \delta(k^{(r)}(0))
\end{cases}
\tag{S40}
$$

661 **(2)+Locality:** $\mathcal{K} = \mathcal{K}_{\text{VEC}}$. , $Q(\mathcal{K}, r) = \{\vec{r} \in B(\mathbb{N}^p, s) : |\vec{r}| = r\}$ *the unit eigenfunctions are*

$$
\begin{cases}
\mathsf{H}_{\text{VEC}}^{(\vec{r})} = \text{span} \left\{ Y_{\vec{r},\vec{l}}(\frac{\cdot}{\sqrt{d_0}}) \right\}_{\vec{l} \in [B(d_0, \vec{r})]} \\
\dim(\mathsf{H}_{\text{VEC}}^{(r)}) \sim ps^{r-1} d_0^r = s^{r-1} d^{1-\alpha+r\alpha} \quad and \quad \lambda(\mathsf{H}_{\text{VEC}}^{(\vec{r})}) \sim p^{-1}(sd_0)^{-r} \delta(k^{(r)}(0))
\end{cases}
\tag{S41}
$$

662 **(3)+Locality + Shifting:** $\mathcal{K} = \mathcal{K}_{\text{GAP}}$. $Q(\mathcal{K}, r) = \{\vec{r} \in B(\mathbb{N}^p, s)/\tau : |\vec{r}| = r\}$, *the unit eigenfunc-*
663 *tions are*

$$
\begin{cases}
\mathsf{H}_{\text{GAP}}^{(\vec{r})} = \text{span} \left\{ \frac{1}{\sqrt{p}} \sum_{\tau \in [p]} Y_{\vec{r},\vec{l}}(\frac{\tau}{\cdot}\sqrt{d_0}) \right\}_{\vec{l} \in [B(d_0, \vec{r})]} \\
\dim(\mathsf{H}_{\text{GAP}}^{(r)}) \sim (sd_0)^r = s^r d^{r\alpha} \quad and \quad \lambda(\mathsf{H}_{\text{GAP}}^{(\vec{r})}) \sim p^{-1}(sd_0)^{-r} \delta(k^{(r)}(0))
\end{cases}
\tag{S42}
$$

664 *Proof.* Our main tool is Theorem D.1.

665 **Base Case $\mathcal{K}_{\text{FCN}}$.** Setting

$$
K(\vec{t}) = k \left( \frac{1}{p} \sum_{j \in [p]} t_j \right)
\tag{S43}
$$

666 and applying Theorem D.1 give

$$
K(\xi^T \eta / d_0) = \sum_{\vec{r} \in \mathbb{N}^p} \hat{K}(\vec{r}) N(d_0, \vec{r})^{-1} \sum_{\vec{l} \in [B(d_0, \vec{r})]} Y_{\vec{r},\vec{l}}(\xi/\sqrt{d_0}) Y_{\vec{r},\vec{l}}(\eta/\sqrt{d_0})
\tag{S44}
$$

667 for $\xi/\sqrt{d_0}$ and $\eta/\sqrt{d_0} \in \mathbb{S}_{d_0-1}^p$. By the chain rule

$$
\begin{aligned}
\hat{K}(\vec{r}) &= \vec{r}!^{-1} \left( K^{(\vec{r})}(0) + \mathcal{O}(\|K\|_{C^{|\vec{r}|+1}(I^p)} p d_0^{-\frac{1}{2}}) \right) \\
&= \vec{r}!^{-1} \left( p^{-\vec{r}} k^{(\vec{r})}(0) + \mathcal{O}(p^{-|\vec{r}|-1} \|k\|_{C^{|\vec{r}|+1}(I)} p d_0^{-\frac{1}{2}}) \right) \\
&= \vec{r}!^{-1} p^{-|\vec{r}|} \left( k^{(\vec{r})}(0) + \mathcal{O}(d_0^{-\frac{1}{2}}) \right)
\end{aligned}
$$

668 As $d_0 \to \infty$, if $k^{(|\vec{r}|)}(0) \neq 0$ then the eigenvalue of the $\vec{r}$-eigenspace is

$$
\lambda(\mathsf{H}^{(\vec{r})}) \sim k^{(|\vec{r}|)}(0) \vec{r}!^{-1} p^{-|\vec{r}|} N(d_0, \vec{r})^{-1} \sim (pd_0)^{-|\vec{r}|} = d^{-|\vec{r}|}
\tag{S45}
$$

669 The dimension is $N(d_0, \vec{r}) \sim d_0^{|\vec{r}|}/\vec{r}!$. This completes the proof of the base case.

670 **+Locality $\mathcal{K}_{\text{VEC}}$.** Recall that

$$
\mathcal{K}_{\text{VEC}}(\xi, \eta) = \frac{1}{p} \sum_{i \in [p]} k \left( \frac{1}{s} \sum_{b \in [s]} \xi_{i+b}^T \eta_{i+b} / d_0 \right),
\tag{S46}
$$

671 which is a sum of kernels supported on patches. Setting

$$
K(t_1, \dots, k_s) = k \left( \frac{1}{s} \sum_{j \in [s]} t_j \right),
\tag{S47}
$$

672 applying Theorem D.1 with $p = s$ to each summand implies

$$
\mathcal{K}_{\text{VEC}}(\xi, \eta) = \frac{1}{p} \sum_{i \in [p]} \sum_{\vec{r} \in \mathbb{N}^s} \hat{K}(\vec{r}) N(d_0, \vec{r})^{-1} \sum_{\vec{l}} Y_{\vec{r},\vec{l}}(\xi_{i:i+s}/\sqrt{d_0}) Y_{\vec{r},\vec{l}}(\eta_{i:i+s}/\sqrt{d_0})
\tag{S48}
$$

$$
= \sum_{\vec{r} \in \mathbb{N}^s} \frac{1}{p} \hat{K}(\vec{r}) N(d_0, \vec{r})^{-1} \sum_{\vec{l}} \sum_{i \in [p]} Y_{\vec{r},\vec{l}}(\xi_{i:i+s}/\sqrt{d_0}) Y_{\vec{r},\vec{l}}(\eta_{i:i+s}/\sqrt{d_0})
\tag{S49}
$$

in which we have applied the Fubini Theorem. Similarly, if $k^{(\vec{r})}(0) \neq 0$, the term

$$\hat{K}(\vec{r})N(d_0, \vec{r})^{-1} \sim k^{(\vec{r})}(0)(sd_0)^{-|\vec{r}|}, \tag{S50}$$

where the $s^{-|\vec{r}|}$ is coming from applying the chain rule to Equation S47. Next, we treat the functions $Y_{\vec{r},\vec{l}}(\xi_{i:i+s}/\sqrt{d_0})$ defined on a patch as functions $Y_{\vec{r},\vec{l}}(\xi/\sqrt{d_0})$ defined on the whole space $(\sqrt{d_0}\mathbb{S}_{d_0-1})^p$ by restricting $\vec{r} \in B(\mathbb{N}^p, s)$. As such we need to count, for a given $\vec{r}$, the number of patches the function $Y_{\vec{r},\vec{l}}(\xi_{i:i+s}/\sqrt{d_0})$ belong to, which turns out to be $(s - \text{diam}(\vec{r}))$. We could reorder the terms in $\mathcal{K}_{\text{VEC}}$ as follows

$$\mathcal{K}_{\text{VEC}}(\xi, \eta) = \sum_{\vec{r} \in B(\mathbb{N}^p, s)} \frac{1}{p} \hat{K}(\vec{r})N(d_0, \vec{r})^{-1}(s - \text{diam}(\vec{r})) \sum_{\vec{l}} Y_{\vec{r},\vec{l}}(\xi/\sqrt{d_0})Y_{\vec{r},\vec{l}}(\eta/\sqrt{d_0}) \tag{S51}$$

Clearly, $Y_{\vec{r},\vec{l}}(\xi/\sqrt{d_0})$ are the eigenfunctions of unit norm with eigenvalues

$$p^{-1}\hat{K}(\vec{r})N(d_0, \vec{r})^{-1}(s - \text{diam}(\vec{r})) \sim p^{-1}k^{(\vec{r})}(0)(sd_0)^{-|\vec{r}|}(s - \text{diam}(\vec{r})) \quad \vec{r} \neq 0, \tag{S52}$$

and $\hat{k}(0)$ when $\vec{r} = 0$.

Note that in the case when the stride is the same as the size of the patch, the $(s - \text{diam}(\vec{r}))$ becomes 1 for all spherical harmonics. As such, smaller strides favor functions with smaller diameters (namely, $\text{diam}(\vec{r})$), breaking the symmetry between functions with small and large diameters.

We turn to compute the dimension of $r$-eigenspace for $r \in \mathbb{N}$. Clearly, for $\vec{r} = 0$ the dimension is 1 and for $|\vec{r}| = 1$ the dimension is $d = pd_0$, which is the dimension of all degree 1 homogenous polynomials. For $|\vec{r}| > 1$, we count the number of spherical harmonics in the 1st patch $\xi_{0:s}$ with $r_0 \neq 0$ and the total number of spherical harmonics in all patches is $p$ time this number. Thus

$$\dim(\mathsf{H}^{(r)}) = p \sum_{\substack{\vec{r} \in \mathbb{N}^s: \\ |\vec{r}|=r, r_0 \neq 0}} N(d_0, \vec{r}) \tag{S53}$$

$$= p \left( \sum_{\substack{\vec{r} \in \mathbb{N}^s: \\ |\vec{r}|=r}} N(d_0, \vec{r}) - \sum_{\substack{\vec{r} \in \mathbb{N}^s: \\ |\vec{r}|=r, r_0 = 0}} N(d_0, \vec{r}) \right) \tag{S54}$$

$$\sim \left( \sum_{\substack{\vec{r} \in \mathbb{N}^s: \\ |\vec{r}|=r}} d_0^r/\vec{r}! - \sum_{\substack{\vec{r} \in \mathbb{N}^{s-1}: \\ |\vec{r}|=r}} d_0^r/\vec{r}! \right) \tag{S55}$$

$$= d_0^r/r!(s^r - (s-1)^r) \sim s^{r-1}d_0^r/(r-1)! \tag{S56}$$

for large $s$.

**+Locality + Pooling** $\mathsf{GAP}_\infty$. The kernel is given by

$$\mathcal{K}_{\text{GAP}}(\xi, \eta) = \frac{1}{p^2} \sum_{i,j \in [p]} k \left( \frac{1}{s} \sum_{b \in [s]} \xi_{i+b}^T \eta_{j+b}/d_0 \right).$$

In what follows we identify $B(\mathbb{N}^p, s)/\tau = B(\mathbb{N}^s, s)$. Applying Theorem D.1 gives

$$\mathcal{K}_{\text{GAP}}(\xi, \eta) = \frac{1}{p^2} \sum_{i,j \in [p]} k \left( \frac{1}{s} \sum_{b \in [s]} \xi_{i+b}^T \eta_{j+b}/d_0 \right),$$

$$= \sum_{\vec{r} \in \mathbb{N}^s} \hat{K}(\vec{r})N(d_0, \vec{r})^{-1} \sum_{\vec{l}} \frac{1}{p^2} \sum_{i,j \in [p]} Y_{\vec{r},\vec{l}}(\xi_{i:i+s}/\sqrt{d_0})Y_{\vec{r},\vec{l}}(\eta_{j:j+s}/\sqrt{d_0})$$

$$= \hat{K}(0)N(d_0, \vec{0}) + \sum_{\vec{r} \in B(\mathbb{N}^p, s)/\tau, \vec{r} \neq 0} \hat{K}(\vec{r})N(d_0, \vec{r})^{-1}\frac{1}{p} \sum_{\vec{l}} Y_{\vec{r},\vec{l}}^\tau(\xi/\sqrt{d_0})Y_{\vec{r},\vec{l}}^\tau(\eta/\sqrt{d_0})$$

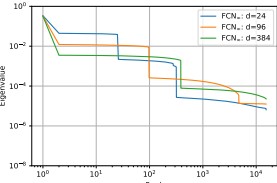 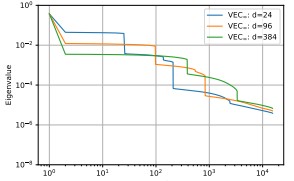 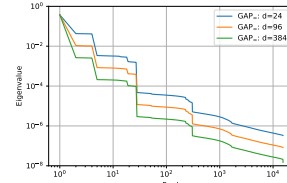

Figure S1: **Eigenvalue Decay of Relu NTK of** $\mathsf{FCN}_\infty$**,** $\mathsf{VEC}_\infty$ **and** $\mathsf{GAP}_\infty$**.** $d_0 = s = 3$. The eigenvalues of $\mathsf{GAP}_\infty$ decays *faster* because with $m = 15k$ many samples, higher order eigenspace can be covered by $\mathsf{GAP}_\infty$ but not by $\mathsf{FCN}_\infty/\mathsf{VEC}_\infty$ as pionted out in Theorem 4.1.

692 where we have defined for $\vec{r} \in B(\mathbb{N}^p, s)/\tau$ with $\vec{r} \neq 0$

$$Y_{\vec{r},l}^{\tau}(\xi/\sqrt{d_0}) = \frac{1}{\sqrt{p}} \sum_{i \in [p]} Y_{\vec{r},l}(\xi_{i:i+s}/\sqrt{d_0}) \tag{S57}$$

693 The eigvenvalue for $\vec{r} = 0$ is $\hat{k}(0)$ and for $\vec{r} \neq 0$ with $k^{(\vec{r})}(0) \neq 0$ are

$$\hat{K}(\vec{r})N(d_0, \vec{r})^{-1}\frac{1}{p} \sim p^{-1}(sd_0)^{-|\vec{r}|}k^{(|\vec{r}|)}(0) \tag{S58}$$

694 Similar to $\mathsf{VEC}_\infty$, the dimension of $r$-eigenspace is $s^{r-1}(d_0)^r/(r-1)!$ for $r \geq 1$. $\qquad\square$

### D.4 Remarks for Theorem D.2

696 The **Baseline** $\mathcal{K}_{\mathsf{FCN}_n}$ is a standard result; see, for example, [17] and [56]. The dimension of $r$-degree
697 harmonic polynomials is $\Theta(d^r)$ and the spectral gap between the 0- and $r$-eigenspaces, namely, the
698 $r$-condition number, $\kappa_r = \Theta(d^r)$. Learning higher order terms (using kernels) in this space suffers
699 from the curse of dimensionality because (1) the number of samples requires to cover a basis of the
700 $r$-eigenspace and (2) the number of gradient steps (or the amount of time for gradient flow) needed to
701 learn the $r$-eigenspace grow with the rate $\Theta(d^r)$. This makes it difficult to learn higher order terms
702 even when $d$ is not very large, e.g., when $r = 4$ and $d = 784$ (Mnist), $d^r \sim 10^{11}$ and lower order
703 terms when $d$ is large, e.g. when $r = 2$ and $d = 3 \times 224^2 \sim 10^5$ (ImageNet), $d^r \sim 10^{10}$.

704 The **+Locality** $\mathcal{K}_{\mathsf{VEC}}$ dramatically reduces both the dimension of the function space and the spectral
705 gap: $\kappa_r \sim \dim(\mathsf{H}^{(r)}) \sim d(sd_0)^{r-1}$. For example, the first layer of ResNet (applied to ImageNet)
706 is a $(7, 7)$ convolution with stride $(2, 2)$ which corresponds to $sd_0 = 7^2 \times 3 \sim d^{0.42}$, where
707 $0.42 \sim \log(7^2 \times 3)/\log(224^2 \times 3)$. With $m \sim d^r$ samples, $\mathcal{K}_{\mathsf{FCN}}$ could cover the $r$-eigenspace,
708 while $K_{\mathsf{VEC}}$ could cover $1 + (r-1)/0.42 \sim (2.4r - 1.4)$-eigenspace.

709 The **+Locality+Pooling** $\mathcal{K}_{\mathsf{GAP}}$. The dimension of the function space is reduced by a factor of $p$ to
710 $\dim(\mathsf{H}^{(r)}) \sim (sd_0)^{r-1}d_0$ and the spectral gap $\kappa_r \sim d(sd_0)^{r-1}$ is unchanged. As a result, $\mathcal{K}_{\mathsf{GAP}}$ is
711 $p$-times more sample-efficient than $\mathcal{K}_{\mathsf{VEC}}$

712 In all cases above, the $r$-condition number $\kappa_r$ can be improved by a factor of $d$ by removing the 0-th
713 eigenspace of the kernels.

### D.5 Proof of Theorem D.1

715 We only need to compute the "Fourier coefficients" $\hat{K}(\vec{r})$. First,

$$\langle P_{\vec{r}}, P_{\vec{r}} \rangle_{L^2(I^p, \omega_{d_0}^p)} = \prod_{j \in [p]} \langle P_{r_j}, P_{r_j} \rangle_{L^2(I, \omega_{d_0})} = N(d_0, \vec{r})^{-1} \left( \frac{|\mathbb{S}_{d_0-1}|}{|\mathbb{S}_{d_0-2}|} \right)^p \tag{S59}$$

716 The last equality could be obtained by applying the addition theorem Lemma 2 and then integrate
717 over $\mathbb{S}_{d_0-1}^p$; see Eq. (4.30) in [55].

718 To handle the numerator in Equation S30, we assume $K$ is sufficiently smooth to avoid the boundary
719 effect. When this is not the case, a little bit effort is needed to handle the boundary values which

will be skipped here. By applying Lemma D.1, integration by parts and continuity of $K^{(\vec{r})}$ on the boundary $\partial I^p$

$$\langle K, P_{\vec{r}} \rangle_{L^2(I^p, \omega_{d_0}^p)} = c_{\vec{r}} \int_{I^p} K(t) \left(\frac{d}{d\vec{t}}\right)^{\vec{r}} \left(1 - \vec{t}^2\right)^{\vec{r} + (d_0 - 3)/2} d\vec{t} \tag{S60}$$

$$= (-1)^{\vec{r}} c_{\vec{r}} \int_{I^p} K^{(\vec{r})}(t) \left(1 - \vec{t}^2\right)^{\vec{r} + (d_0 - 3)/2} d\vec{t} \tag{S61}$$

$$= (-1)^{\vec{r}} c_{\vec{r}} (\mathcal{M}(K, d_0) + \epsilon(K, d_0)) \tag{S62}$$

where $K^{(\vec{r})}$ is the $\vec{r}$ derivative of $K$, the coefficient is given by Lemma D.1 and

$$c_{\vec{r}} = \prod_{j \in [p]} c_{r_j} = \prod_{j \in [p]} \frac{(-1)^{r_j}}{2^{r_j}(r_j + (d_0 - 3)/2)_{r_j}} \sim \prod_{j \in [p]} (-1)^{r_j} d_0^{-r_j} = (-1)^{\vec{r}} d_0^{-\vec{r}} \tag{S63}$$

and the major and error terms are given by

$$\mathcal{M}(K, d_0) = K^{(\vec{r})}(0) \int_{I^p} \left(1 - \vec{t}^2\right)^{\vec{r} + (d_0 - 3)/2} d\vec{t} = K^{(\vec{r})}(0) \prod_{j \in [p]} \frac{|\mathbb{S}_{2r_j + d_0 - 1}|}{|\mathbb{S}_{2r_j + d_0 - 2}|} \tag{S64}$$

$$\epsilon(K, d_0) = \int_{I^p} (K^{(\vec{r})}(t) - K^{(\vec{r})}(0)) \left(1 - \vec{t}^2\right)^{\vec{r} + (d_0 - 3)/2} d\vec{t} \tag{S65}$$

For the error term, we use the mean value theorem to bound

$$|(K^{(\vec{r})}(t) - K^{(\vec{r})}(0))| \leq \|K\|_{C^{|\vec{r}|+1}(I^p)} \sum_{j \in [p]} |t_j| \tag{S66}$$

and

$$|\epsilon(K, d_0)| \leq \|K\|_{C^{|\vec{r}|+1}(I^p)} \int_{I^p} \left(1 - \vec{t}^2\right)^{\vec{r} + (d_0 - 3)/2} d\vec{t} \sum_{j \in [p]} \left( \frac{\int_I |t_j| \left(1 - t_j^2\right)^{r_j + (d_0 - 3)/2} dt_j}{\int_I \left(1 - t_j^2\right)^{r_j + (d_0 - 3)/2} dt_j} \right) \tag{S67}$$

$$\sim \|K\|_{C^{|\vec{r}|+1}(I^p)} \prod_{j \in [p]} \frac{|\mathbb{S}_{2r_j + d_0 - 1}|}{|\mathbb{S}_{2r_j + d_0 - 2}|} \sum_{j \in [p]} d_0^{-1} \left( \frac{|\mathbb{S}_{2r_j + d_0 - 1}|}{|\mathbb{S}_{2r_j + d_0 - 2}|} \right)^{-1} . \tag{S68}$$

Since for any $\alpha \in \mathbb{N}$, as $d_0 \to \infty$,

$$\frac{|\mathbb{S}_{\alpha + d_0 - 1}|}{|\mathbb{S}_{\alpha + d_0 - 2}|} = \pi^{\frac{1}{2}} \Gamma((\alpha + d_0 - 1)/2)/\Gamma((\alpha + d_0)/2) \sim \pi^{\frac{1}{2}} (d_0/2)^{-\frac{1}{2}} \tag{S69}$$

We have

$$|\epsilon(K, d_0)| \lesssim \|K\|_{C^{|\vec{r}|+1}(I^p)} p d_0^{-\frac{1}{2}} \prod_{j \in [p]} \frac{|\mathbb{S}_{2r_j + d_0 - 1}|}{|\mathbb{S}_{2r_j + d_0 - 2}|} \tag{S70}$$

Therefore

$$\langle K, P_{\vec{r}} \rangle_{L^2(I^p, \omega_{d_0}^p)} = c_{\vec{r}} \left( K^{(\vec{r})}(0) + \mathcal{O}(\|K\|_{C^{|\vec{r}|+1}(I^p)} p d_0^{-\frac{1}{2}}) \right) \prod_{j \in [p]} \frac{|\mathbb{S}_{2r_j + d_0 - 1}|}{|\mathbb{S}_{2r_j + d_0 - 2}|} \tag{S71}$$

Plugging back to Equation S30, we have

$$\hat{K}(\vec{r}) = (-1)^{\vec{r}} c_{\vec{r}} N(d_0, \vec{r}) \left( K^{(\vec{r})}(0) + \mathcal{O}(\|K\|_{C^{|\vec{r}|+1}(I^p)} p d_0^{-\frac{1}{2}}) \right) \left( \prod_{j \in [p]} \frac{|\mathbb{S}_{2r_j + d_0 - 1}|}{|\mathbb{S}_{2r_j + d_0 - 2}|} \right) \left( \frac{|\mathbb{S}_{d_0 - 1}|}{|\mathbb{S}_{d_0 - 2}|} \right)^{-p} \tag{S72}$$

Since, for $\vec{r}$ and as $d_0 \to \infty$

$$\frac{c_{\vec{r}}}{(-1)^{\vec{r}} d_0^{-\vec{r}}} \to 1 \quad \text{and} \quad \frac{N(d_0, \vec{r})}{d_0^{|\vec{r}|}/r!} \to 1 \quad \text{and} \quad \left( \prod_{j \in [p]} \frac{|\mathbb{S}_{2r_j + d_0 - 1}|}{|\mathbb{S}_{2r_j + d_0 - 2}|} \right) \left( \frac{|\mathbb{S}_{d_0 - 1}|}{|\mathbb{S}_{d_0 - 2}|} \right)^{-p} \to 1 \tag{S73}$$

and thus

$$\hat{K}(\vec{r}) = r!^{-1} \left( K^{(\vec{r})}(0) + \mathcal{O}(\|K\|_{C^{|\vec{r}|+1}(I^p)} p d_0^{-\frac{1}{2}}) \right) \tag{S74}$$

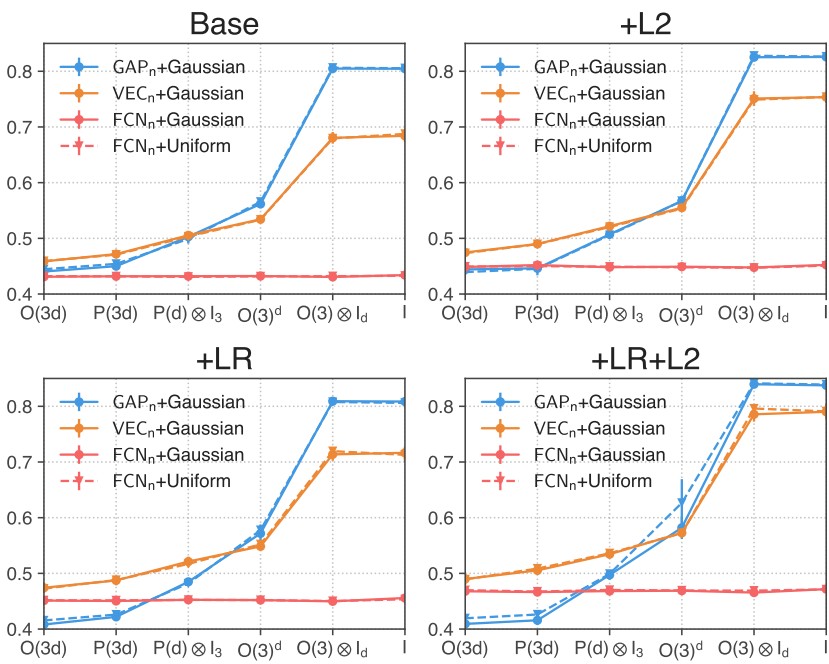

Figure S2: **Replacing the Gaussian initialization by uniform distribution does not change the performance much.**

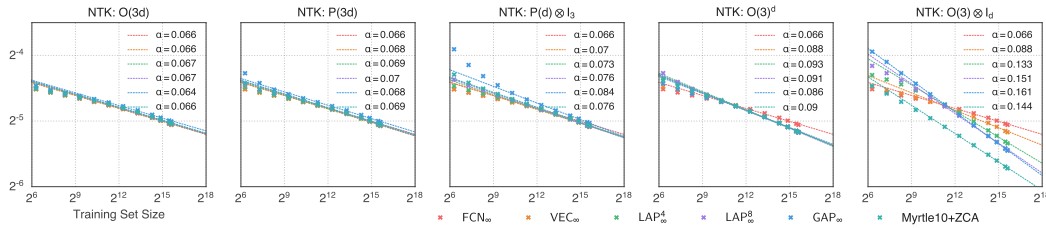

Figure S3: **Scaling Law of Infinite Network vs Different Symmetries.**

## E    Plots Dump

### E.1    Gaussian vs Uniform Initialization

### E.2    Scaling Law for Infinite Networks

### E.3    Finite Width Effect of $\mathsf{VEC}_n$.

### E.4    Implication of Theorem D.2

We investigate the data-efficiency of various architectures on various tasks. The tasks are to learn harmonic polynomials containing degree $r = 1, 24$ in $(\mathbb{S}_2)^{16}$. The MSE of each degree is normalized to be $0.5$ and the MSE of the zero predictor is $1.5$. There are 5 types of polynomials/tasks (columns in Fig.S5):

1. **Non-local**, which is our baseline, corresponding to generic polynomials without structure information. The optimal kernel to solve this task in this paper is $\mathcal{K}_{\mathsf{FCN}}$.

2. **Non-local+shift**, adding shifting invariance to **Non-local**. The optimal kernel is $\mathcal{K}_{\mathsf{FCN}}$ + Shifting invariance.

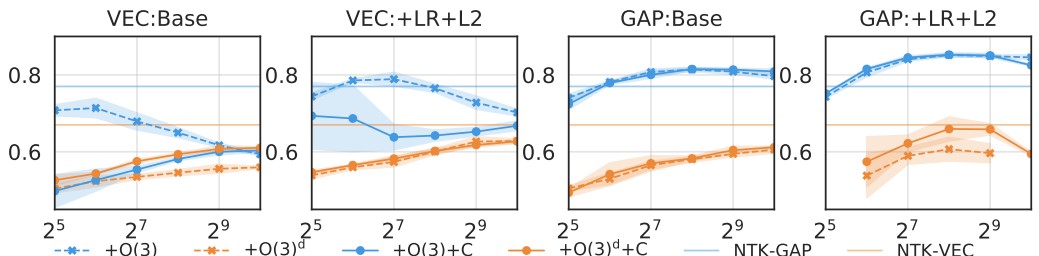

Figure S4: **Performance vs Width for** $\mathsf{VEC}_n$ **and** $\mathsf{GAP}_n$ With the $O(3)^d$ symmetry imposed on the system, performance of $\mathsf{VEC}_n$ is below the performance of $\mathsf{VEC}_\infty$ (67%), but monotonically improves as the width $n$ increases. However, with the original coordinate system ($O(3) \otimes \mathbf{I}_d$), performance (without centering) improves and then degrades significantly after the peak. This is because the network is less sensitive to the $O(3)$ symmetry. In stark contrast, the performance of $\mathsf{GAP}_n$ improves from $n = 32$ to $n = 512$ but only slightly degrades at $n = 1024$. With and without centering, the performance of $\mathsf{GAP}_n$ is similar while the performance of VEC is dramatically different.

3. **Local**: the polynomial depends locally on patches of size (3, 3), i.e. $\mathbb{S}_2^3$; The optimal kernel is $\mathcal{K}_{\mathsf{VEC}} = \mathcal{K}_{\mathsf{FCN}}$.

4. **Local + Sparse**: the polynomial depends only on one *single* patch. The optimal kernel should be a FCN-kernel defined on that patch, which is a not available among our kernels. The $\mathcal{K}_{\mathsf{VEC}} = \mathcal{K}_{\mathsf{FCN}}$ is the second best.

5. **Local + Shift**: enforcing shifting invariance **Local**. The optimal kernel is $\mathcal{K}_{\mathsf{GAP}}$.

In the (5, 5)-panel Fig.S5, we plot the MSE ($y$-axis) vs $\log(m)/\log(d)$ ($x$-axis), where $m$ is the number of samples and $d = 3 * 16 = 48$ is the dimension of the input data, for different learning algorithms: (1) NNGP, the Gaussian Process kernel (2) NTK, the kernel of infinite width network corresponding to training only the first layer (3) NN, finite width networks with width $n = 16$, (4) $n = 4096$ and (5) $n$=best, which is obtained as follows: for each $m$, we sweep over $n = 16 \rightarrow 4096$ dyadically by a factor of 2 and report the best performance.

For Non-FCN kernels, we choose $m$ up to $5120 \times 4 \sim 20k$, since the MSE have already reached a very small number, i.e. learning all frequencies $r = 1, 2, 4$. For FCN, we choose $m$ up to $5120 \times 32 \sim 160k$, the biggest $m \times m$ matrix that we could be solved within our compute budget. However this still falls in short with $d^4 = (48)^4 \sim 5000k = 5 \times 10^6$, the dimension of 4-eigenspace. Not surprising, the vanilla FCN kernel could not learn the $r = 4$ frequency for all tasks (first row). However, FCN kernel + Shifting could learn **Non-local+shift** and **Local + Shift** with $m \sim d^3$, since the symmetry *shifting* reduces the dimension of $r$-eigenspace by a factor of $d$.

Finite width $\mathsf{FCN}_n$ does better than kernels when learning (higher) $r = 4$ frequency, requiring $m \sim d^{3+}$ many samples (first row of the plot), while kernel would require $d^4$ many samples. It does even better on the task **Local + Sparse** with smaller $n$ and equally less good in **Non-local, Non-local + shift, Local** and **Local + Shift**. This says finite width networks are good at handling *sparsity* but not *locality*, which has to be imposed by human into them as a form of inductive biases.

Now let us focus on the third row $\mathsf{LCN}_n$. Not surprising, it does bad on the first two tasks **Non-local** and **Non $-$ local $+$ shift** because the function space is to small. For the remaining tasks, kernels and finite width networks are efficient and competitively with each other. Only in the task **Local $+$ Sparse** $\mathsf{LCN}_n$ does noticeable better than kernel, demonstrating the strong ability of finite width networks in handling sparsity.

With weight-sharing (4th-row), VEC does noticeably better in all tasks that require locality. It is an interesting direction to understand the analytic reason behind it.

With the correct prior, the $\mathsf{GAP}_\infty$ does equally well as $\mathsf{GAP}_n$. Both of them are the most data-efficient among all other architectures/algorithms in the plot when handling the task **Local $+$ Shift**.

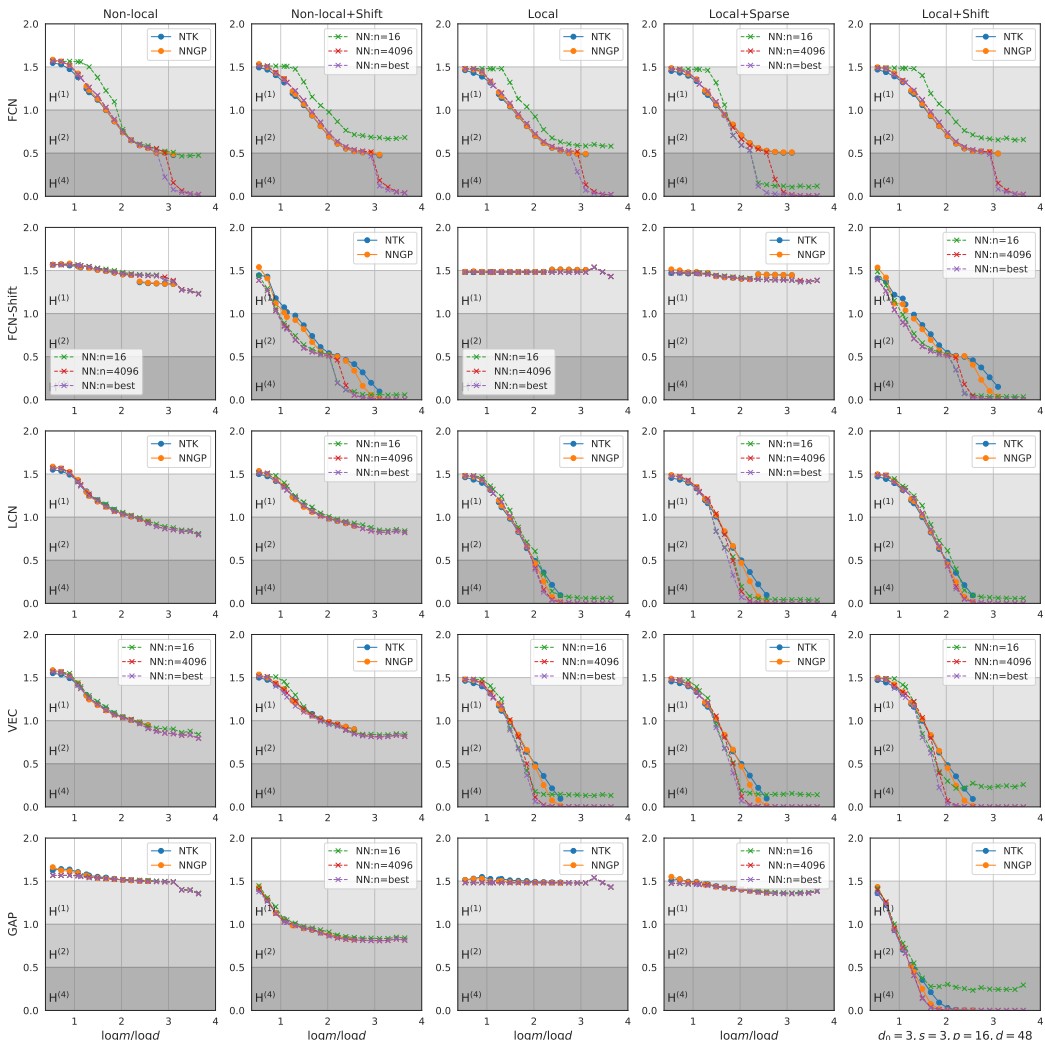

Figure S5: **Impact of Locality and Symmetries.** Performance of 5 types of kernels and finite width networks on 5 types of tasks.

### E.5 Scaling Plots for ResNet34 and ResNet101

### E.6 ImageNet Samples

## F An example for invariance

**Example 1.** *Linear Regression: Let $\mathcal{A}$ be the (determnistic) algorithm that outputs the minimum norm linear regrssion solution, then $\mathcal{A}$ is $O(3d)$-invariant, because for $\mathcal{D}_T = (\mathcal{X}_T, \mathcal{Y}_T)$, the prediction*

$$\mathcal{A}(\mathcal{D}_T)(\mathcal{X}_*) \equiv \mathcal{X}_* \mathcal{X}_T^T (\mathcal{X}_T \mathcal{X}_T^T)^\dagger \mathcal{Y}_T = \tau \mathcal{X}_* (\tau \mathcal{X}_T)^T ((\tau \mathcal{X}_T)(\tau \mathcal{X}_T)^T)^\dagger \mathcal{Y} \equiv \mathcal{A}^\tau(\mathcal{D}_T)(\mathcal{X}_*), \quad \text{(S75)}$$

*where $\tau x \equiv x U_\tau$, here $x$ is a row vector and $U_\tau \in O(3d)$ is the matrix representation of $\tau$.*

*If $\mathcal{A}$ is the (stochastic) algorithm that applies gradient flow to solve the linear regression $\mathcal{X}_T \omega = \mathcal{Y}_T$ with the MSE loss and the entries of $\omega$ are initialized with iid standard Gaussian, then each $f \in \mathcal{A}(\mathcal{D}_T)$ is a draw from the posterior, namely,*

$$f(\mathcal{X}_*) \sim \mathcal{N}\left(\mathcal{X}_* \mathcal{X}_T^T (\mathcal{X}_T \mathcal{X}_T^T)^\dagger \mathcal{Y}, \mathcal{X}_* \mathcal{X}_*^T - \mathcal{X}_T \mathcal{X}_T^T (\mathcal{X}_T \mathcal{X}_T^T)^\dagger \mathcal{X}_T^T \mathcal{X}_*\right). \quad \text{(S76)}$$

*Note that the distribution is invariant to coordinate rotation by any $\tau \in O(3d)$ and therefore $(\tau \mathcal{D}, \mathcal{M}, \mathcal{I}) = (\mathcal{D}, \mathcal{M}, \mathcal{I})$ for all $\tau \in O(3d)$.*

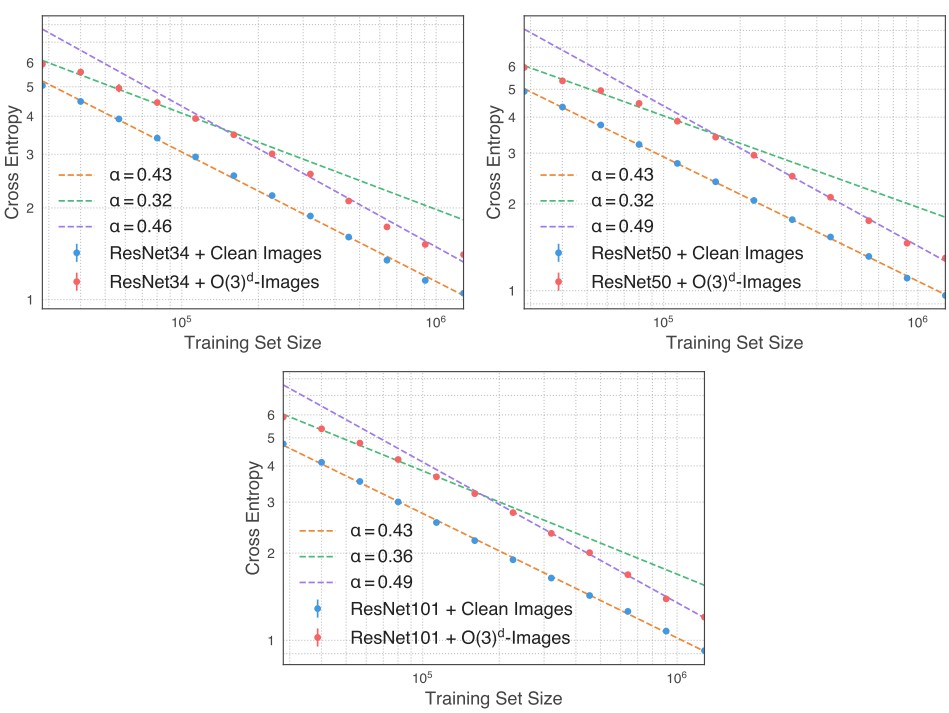

Figure S6: **Scaling vs Rotation**

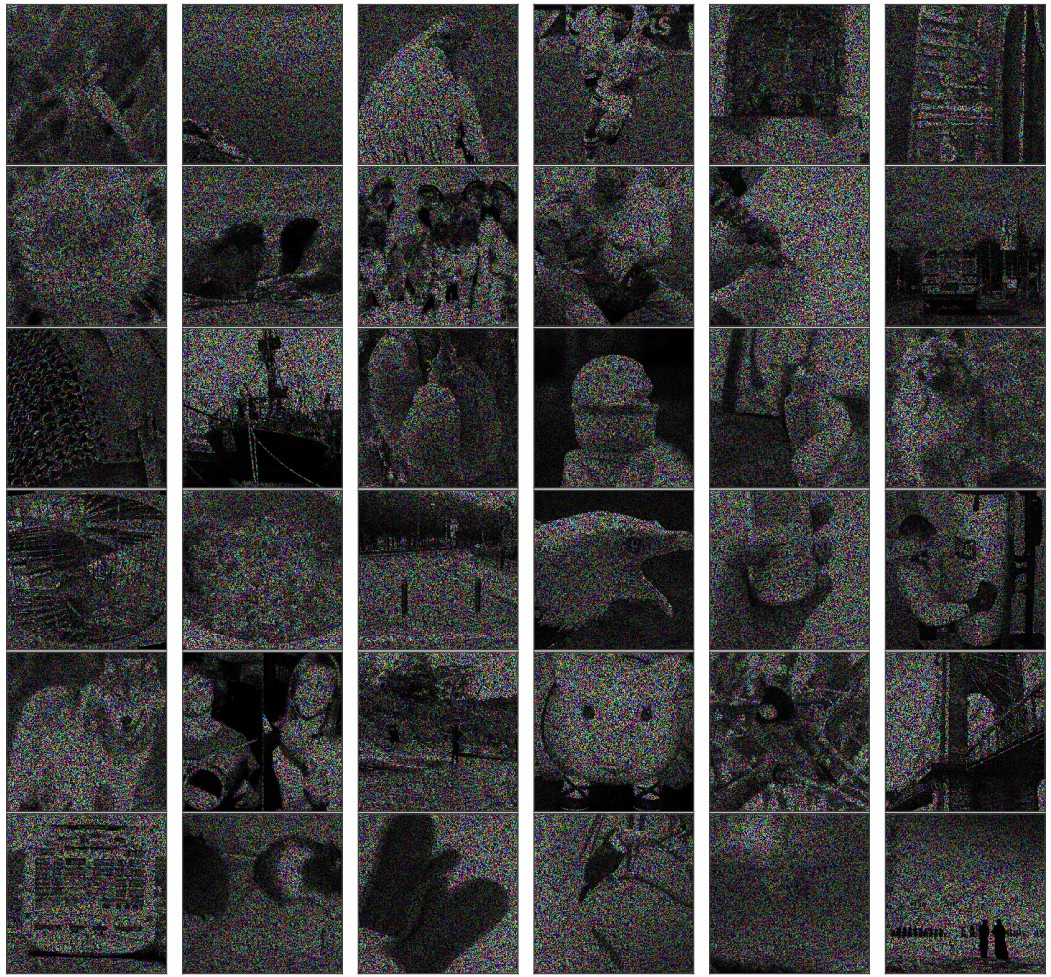

Figure S7: $O(3)^d-$**Rotated ImageNet Samples. Seed=1**

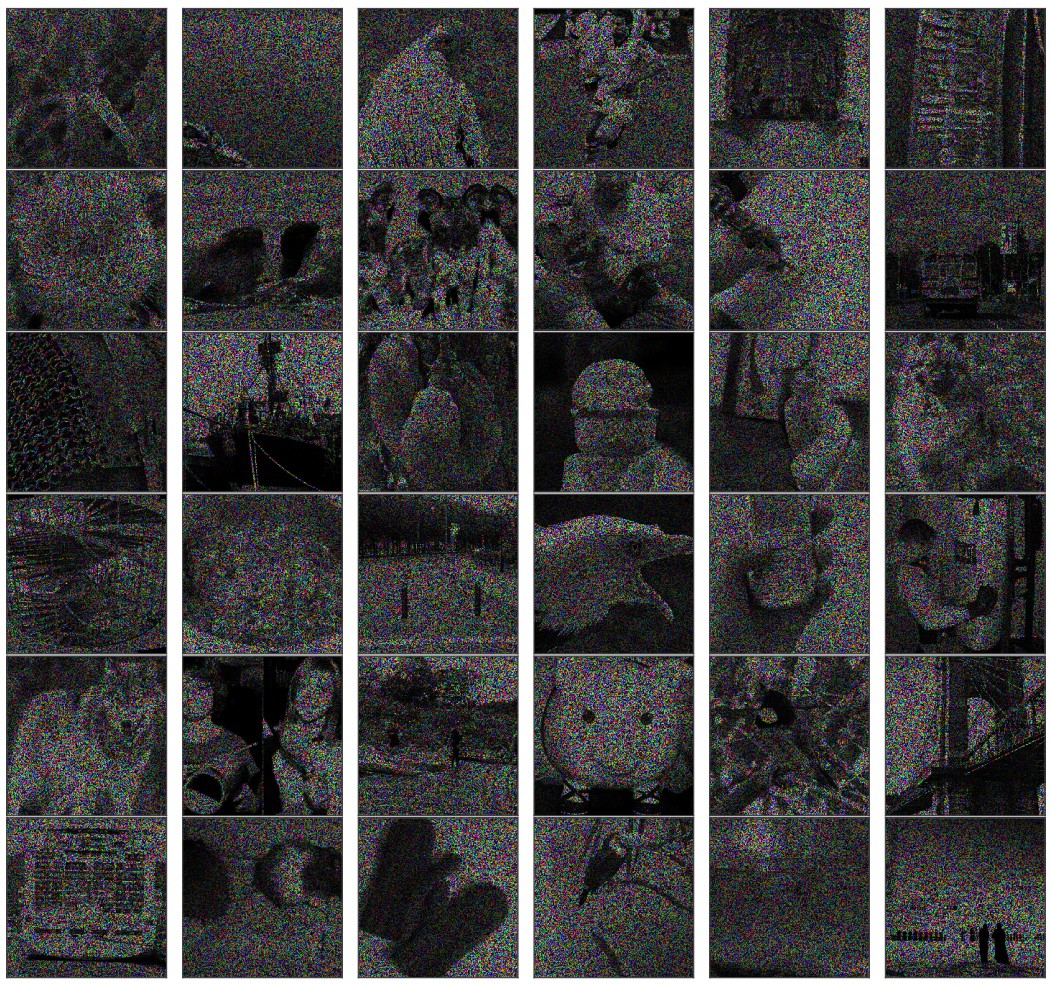

Figure S8: $O(3)^d-$**Rotated ImageNet Samples. Seed=2**

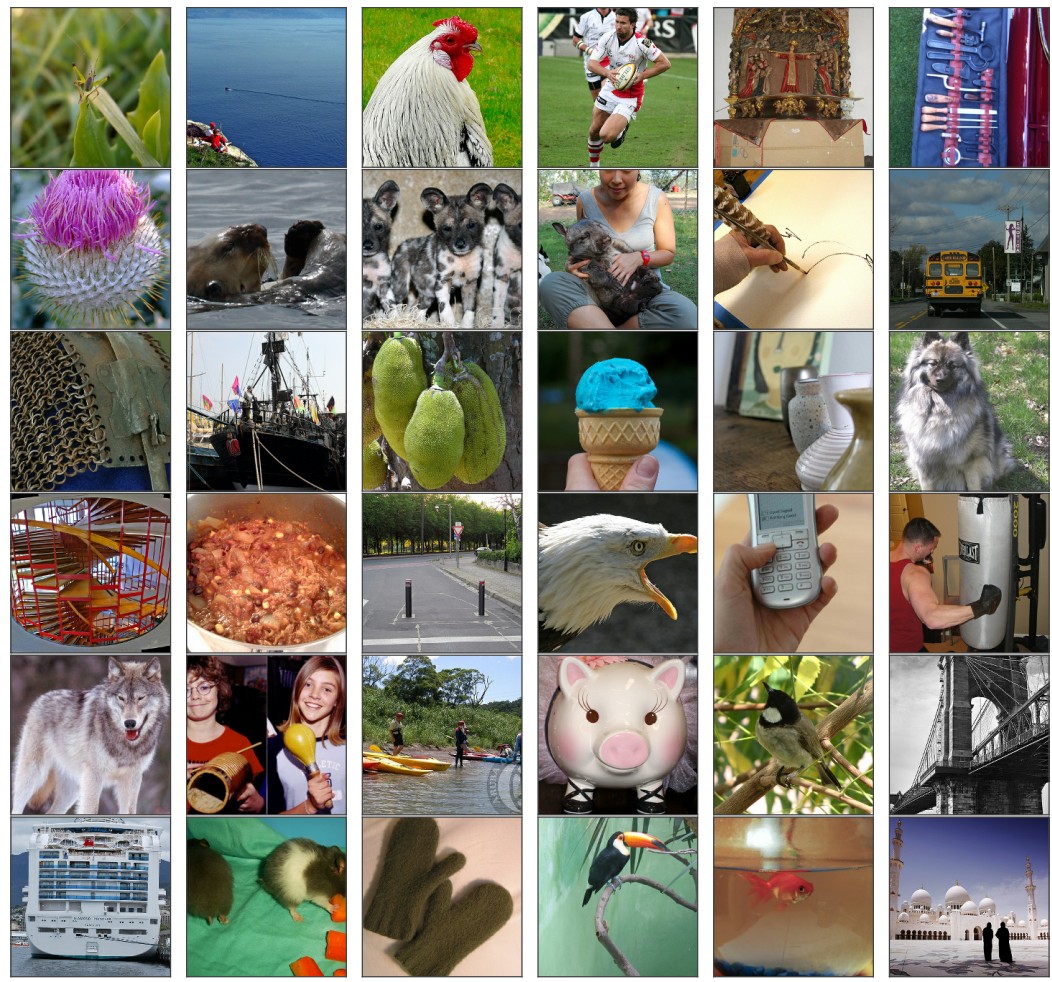

Figure S9: **Clean ImageNet Samples**