# OpenReview forum: "What Breaks The Curse of Dimensionality in Deep Learning?"
_NeurIPS.cc/2021/Conference — NeurIPS 2021 Submitted_

### Official Review · Reviewer_dgpq · 2021-07-14

**Rating:** 6
**Confidence:** 3

**Summary:**

The paper studies the symmetries of the the system (Arch, Data, Algorithm) and provides extensive empirical results. the paperis well-written and flows nicely.

**Limitations And Societal Impact:**

Yes

**Main Review:**


Advantages:

- Understanding symmetry in Deep Learning is still a major open question. This paper introduces a new framework to characterize this symmetry by considering intergated (arch, data, alg) systems.

- The empirical results support the claimed symmetries although (the symmetries) they're not very accurate in some scenarios.

- The paper is well-written


Weaknesses:

- The bulk of the analysis relies on the NTK and NTK parameterization. It is now well known that the NTK regimes does not capture what happens in deep neural networks (see, e.g. Yang et al. Feature Learning in Infinite-Width Neural Networks). Does the analysis extend to the standard parameterization (i.e. without the 1/sqrt{n})?

**Time Spent Reviewing:**

3 hours

---

> ### Author Response · Authors · 2021-08-10
> **Response to Reviewer dgpq**
>
> We thank the reviewer for many constructive feedbacks. The reviewer pointed out one weakness of our paper, which we respectfully disagree with. Please see the detailed response below.
>
> >Weaknesses:
> The bulk of the analysis relies on the NTK and NTK parameterization. It is now well known that the NTK regime does not capture what happens in deep neural networks (see, e.g. Yang et al. Feature Learning in Infinite-Width Neural Networks). Does the analysis extend to the standard parameterization (i.e. without the 1/sqrt{n})?
>
>
> Thanks for the question. To clarify, most of the results (in particular, the ones about symmetry) about finite width networks are independent from the parameterization methods (i.e. NTK vs Standard parameterization vs MeanField/Feature Learning parameterization) and only the part that involves the neural tangent kernels requires the NTK parameterization. More precisely, the symmetries of the (D, M, I) system depends on the architecture, initialization methods (Gaussian vs Uniform), optimization algorithm and etc, but not on the parameterization method (i.e. NTK vs standard vs MeanField). That is to say, the system with any parameterization methods for FCNs (with Guassian initialization) still suffers from the spurious symmetry (O(3d)) and the feature learning regime won’t be able to fix it. As a corollary, FCN-based analysis in the so-called feature learning limit won’t be able to explain the success of deep learning in image classification.
>
> >It is now well known that the NTK regime does not capture what happens in deep neural networks.
>
> While we agree that the NTK regime does not tell the whole story, we believe it does contribute significantly in terms of the inductive bias of the model. Broadly speaking, in addition to the benefits from the huge amount of data, deep learning also succeeds in large part due to two types of biases: hand-designed inductive biases (e.g. from architectures), and implicit biases (e.g. from SGD). Quantitative understanding of the benefits of each of them is of fundamental importance and is still largely open. Our theoretical results make some decent progress in the understanding of the inductive biases. We readily admit that the NTK regime cannot capture key aspects of implicit biases; however, our results show that the NTK regime does capture certain key aspects of the inductive biases, e.g. locality and shifting invariance. Our theoretical results (eigen-decomposition) already demonstrate a dramatic gain ($d^r$ v.s. $d_0^r$) from the inductive biases alone (+locality +symmetries). That is to say, certain benefits of the inductive biases are not unique to the feature learning regime and persist even under NTK dynamics -- we don’t have to dive into the either the intractable mean-field dynamics (https://arxiv.org/abs/1804.06561) or unwieldy recursive formulas (Yang et al. Feature Learning in Infinite-Width Neural Networks) to understand them. Isn’t that great news?
>
> In addition, we are not aware of any existing work that is able to demonstrate such a huge gain from GAP/VEC against FCN in the feature learning regime. Moreover, our empirical results suggest that the inductive biases might be the first order contribution to the performance of neural networks, while the implicit biases (feature learning) are secondary. E.g. in the CIFAR10 experiments (figure 1), infinite width GAP (NTK regression) obtains 78% test accuracy without data augmentation while finite width FCN with aggressive data augmentation (Mixup) can only achieve ~70%. In a more controlled experiment setting with synthetic data (Fig S5 last column), infinite width VEC/GAP (+locality, + shifting invariance) always outperform any finite width FCN with SGD by a large margin. Therefore, our theoretical and empirical results highlight the significance of locality, which has largely been overlooked by the research community.
>
> As such, being able to explain a complex phenomenon in a simple (NTK vs feature learning regime) and essential manner should be considered as a strength of paper rather than a weakness.
>
> We hope the reviewer could consider increasing the score if we have addressed the concern.

---

> > ### Comment · Reviewer_dgpq · 2021-08-25
> > **feedback**
> >
> > Thank you for your detailed feedback.
> >
> > When you define the model, you use the NTK parameterization (which is not standard). If your results extend to standard parameterization, I suggest you make that clear in section 2.1.
> >
> > Although the NTK regime might be interesting, it does not always capture the inductive bias as you suggested. If you consider an infinite width neural network of depth L, then as you increase L, the NTK converges to a trivial constant kernel (in general). Empirically, the gap in performance between finite width nets and their infinite width NTK regimes generally increases w.r.t depth L. This suggests that NTK regime does not capture the inductive bias of deep nets.

---

> > > ### Author Response · Authors · 2021-08-25
> > > **reply to feedback**
> > >
> > > Thanks for your feedback!
> > >
> > > >When you define the model, you use the NTK parameterization (which is not standard). If your results extend to standard parameterization, I suggest you make that clear in section 2.1.
> > >
> > > Thanks for the suggestion. Indeed, we already have that in Remark S2 in SM  (see Page2 in SM). We agree that we should have made it clear in the main text.
> > >
> > > Please see the response below.
> > >
> > > >Although the NTK regime might be interesting, it does not always capture the inductive biases as you suggested. If you consider an infinite width neural network of depth L, then as you increase L, the NTK converges to a trivial constant kernel (in general).
> > >
> > > We agree that NTK may not cover the inductive biases of **all** networks, in particular when the networks are *degenerate* / *ill-posed*. The example you provided (width $n\to\infty$ first then $L\to\infty$) is indeed a *degenerate* case for both **Finite** width network and NTK regime, as the reviewer pointed out that `the NTK converges to a trivial constant kernel`. This indicates the **finite** width network is also **non-trainable** or **non-generalizable** (see [1]). We are not aware of any existing work that is able to train a large finite width large depth network to perform reasonably well when the NTK is degenerate. The closest one that we can find is [2] (training 10,000 deep CNN). One key idea there to achieve this goal is indeed forcing the NTK away from the constant kernel. In sum, given that such finite width network (with almost constant NTK) performs very badly, it may not be reasonable to expect the NTK to capture meaningful inductive biases.
> > >
> > > > Empirically, the gap in performance between finite width nets and their infinite width NTK regimes generally increases w.r.t depth L.
> > >
> > > We thank the reviewer for this comment. Questions like (1) when NTK (in general kernels) is better than finite width networks, or vice versa, (2) how much is one better than the other are still open questions in the field. Note that the NTK can outperform finite width networks (see Fig1 in [3] for FCN and CNN-VEC), and vice versa. In [4], the authors show that infinite width networks outperform finite-width counterpart in small-data tasks.  The phenomena you mention/observe may due to **implicit biases* and/or data-augmentation rather than **inductive biases**. Could you provide a pointer to this statement so that we can dig into the experimental details? Please also see the test accuracy of Myrtle5/7/10 below, in which the gap changes between kernels and finite width networks are (from 5 ->7->10 layers) `4.0% -> 3.6% -> 3.7%`.
> > >
> > > >This suggests that NTK regime does not capture the inductive bias of deep nets.
> > >
> > > We respectfully disagree with the reviewer about this point. Empirically, increasing the depth does improve performance of NTK. Here are the test accuracy of Cifar10 (from table 3 in [5]) using Myrtle5/7/10 kernels and the corresponding finite width networks:  `85.8/86.6/87.5` (kernel) and `89.8/90.2/91.2` (finite width SGD) resp. The gains from 5->7 and 7->10 are `0.8/0.9` and `0.4/1.` for NTKs and finite width networks, resp. It seems, at least in this setting, the performance gain due to depth for finite width networks is mostly captured by the NTKs.  Theoretically, we can prove (work in progress) the NTKs can indeed capture certain inductive biases of deep nets, more precisely, the benefits of hierarchical locality.
> > >
> > > Overall, we agree with the reviewer that NTK (more general, kernel methods) has its own limitation. However, we must be open-minded to all kinds of methods: on one hand we should push the theory of NTK (kernels in general) forward to address core ML questions that are addressable by this framework, e.g. double descent phenomena, inductive biases and on the other hand, we should keep developing new tools to tackle those questions that are unaddressable by NTK/kernels, e.g. "feature learning", evolution of NTK, etc.
> > >
> > > Please let us know if you have any other questions and again, thanks for your insightful feedback.
> > >
> > > [1] Disentangling Trainability and Generalization in Deep Neural Networks
> > >
> > > [2] Dynamical Isometry and a Mean Field Theory of CNNs: How to Train 10,000-Layer Vanilla Convolutional Neural Networks
> > >
> > > [3] Finite Versus Infinite Neural Networks: an Empirical Study
> > >
> > > [4] Harnessing the Power of Infinitely Wide Deep Nets on Small-data Tasks
> > >
> > > [5] Neural Kernels Without Tangents

---

### Official Review · Reviewer_XBfK · 2021-07-14

**Rating:** 5
**Confidence:** 3

**Summary:**

This paper tries to understand why a convolutional neural network and its variants with a proper training algorithm can learn to recognize images with only limited training data. Theoretically, by considering the data, model, and inference algorithm as a system, the authors study the symmetries of several different systems, and compute and analyze the eigendecomposition of their corresponding one-hidden-layer infinite-width kernels on spherical input space. The authors also did experiments to study how the symmetries affect the performance of different models on different data and validate their theoretical results on eigendecomposition.

**Limitations And Societal Impact:**

The authors briefly discussed the limitations of their work at the end of this paper, and it may be better to talk more about them. I do not see any immediate negative societal impacts of this paper.

**Main Review:**

Originality: It is interesting to study the data, model, and training algorithm as an entire system and analyze its symmetries, which differs from previous research. The related works appear to be adequately cited.

Quality: The theoretical results in this paper seem to be correct, and they are supported by the experimental results.

Significance:

- The justification for analyzing the symmetry of the data-model-algorithm system seems unclear. This is my major concern about this paper. The authors theoretically showed that different systems have different basic symmetries, and empirically demonstrated that changing the symmetries can affect the performances of the trained models. However, the reason why this change of symmetry is the cause for the change in performances is unclear. The authors did not seem to provide strong theoretical evidence or insights explaining the relationship between symmetry and performance. Besides, in the experiments, changing the symmetries implicitly changes a lot of other factors. For instance, an image with each pixel "rotated" on the channel space is no longer a valid image. It might be better if the authors could provide more theoretical insights about the relationship between symmetry and performance so the experimental results can be better justified.

- Studying the data-model-algorithm system as a whole seems to make the analysis in this paper more complicated than directly analyzing the model structure or the data. The symmetry in this paper is defined on the entire system, but this property mainly comes from the symmetries in the model. It also depends on the distribution used for initialization and the training algorithm, but these are almost fixed in reality. Besides, when doing experiments, the authors are also changing one variable at a time. Therefore, it seems unclear why the authors want to analyze the entire system, which is a more complicated object than individual components.

- The authors computed the eigendecomposition of different models' one-hidden-layer infinite-width kernels on a particular input space and validated their results by experiments, but this setting might be too specific, making the results somewhat restricted.

Clarity:

- This paper might contain too many claims, and there seems to be a lack of focus. It seems from the abstract and the conclusion section that the authors want to emphasize the important role of locality in helping neural networks escape the curse of dimensionality. However, in the introduction section, especially the contribution part, the authors are making a lot of claims and listing many results, which may confuse the reader. It might be better to shorten the introduction and perhaps re-organize these parts to focus more on the main claim of the paper. Besides, at the end of section 3 where the title is about symmetries, the authors are talking about DIDE, which may be only related to the training data. Moving this part out as a separate section may improve the flow of this paper.

- Some of the experimental details, e.g., the hyperparameter settings, are not mentioned in this paper, which makes their experimental results not convincing enough.

- There are some other parts of the paper that may need minor improvements, and the details will be provided in the "Minor Comments" section below.

Minor Comments:

- In line 85, the authors claimed that "we prove the opposite is true". What they actually prove is that the function class of $LCN_n$ is a subset of that of $VEC_{dn}$, but the direct opposite seems to be $LCN_n$ versus $VEC_n$. Therefore, this claim may seem a bit misleading.

- For section 2.1, the authors provide many formulas for different network models, and it might be better to also have some figures for the network models so the readers can better understand them.

- In line 190, the notation "$=^d$" is not defined.

- In Figure 1, the y-axis is not defined, and the second subfigure has no legend.

- In the experiments in Figure 1 and Figure 3, the authors did not mention the stopping criteria for the training process and it is unclear whether they run each experiment multiple times. They also did not mention why they choose the experimental settings NN/NN+/NN++. These may make their empirical results less convincing.

- Section 4 ends with a theorem and the implications of this theorem are in the appendix. It might be better to summarize the implications in the main text to improve the flow of this paper.

Typos:

- Equation 2: "$\sum_{j=1}^{n^l}$" -> "$\sum_{i=1}^{n^l}$"
- Equation 3: "$\sum_{\alpha\in[d]}$" -> "$\sum_{\alpha\in[d],i\in[n]}$"
- Line 167: "where is the infinite width limit of $\Theta$" -> "where $\Theta$ is the infinite width limit of $\hat{\Theta}$"
- Line 262, 271, 279: "Fig. S2" -> "Fig. 4"
- Line 270: "which" -> "which is"
- Line 274: "the dataset grows, ...... as that of" -> "as the dataset grows, ...... that of"

---------------------Update--------------------------

I have read all the other reviews and the authors' responses, and I have decided to increase my score by 1, i.e., I think it's on the borderline and slightly leaning towards rejection. The detailed reasons for this are provided below:

- After reading the authors' response, the justification for analyzing the symmetries of the data-model-algorithm system becomes more clear to me. Thank the authors for providing examples explaining the role of symmetries in the system, and I agree with the authors that understanding these symmetries is an important step towards understanding the empirical success of convolutional neural networks. It might also be better if the authors could explain more about this in their paper to help the readers better understand the importance of this topic.

- My current major concern is the clarity of this paper, which is the main reason why I am still tending to reject this paper. On the one hand, this paper seems to have a lack of focus. The authors included a lot of content in this paper, but the structure and organization of these contents might need some improvements so that the readers can better understand the core idea of this paper. As I suggested in my original review, a more concise and clear introduction might help a lot. On the other hand, as also suggested by the other reviewers, there are many things that need further clarifications, e.g., the insights and consequences of Theorem 4.1, the "Conjecture/Ideas of DIDE", more detailed experimental settings and explanations, and comparisons to some prior works. These modifications may require a major revision of this paper. Therefore, the current form of this paper is perhaps not ready for publication.

**Time Spent Reviewing:**

10

---

> ### Author Response · Authors · 2021-08-10
> **Respond to XBfK**
>
> We thank the reviewers detailed and constructive feedbacks, which will certainly help improve the readability of the paper. Here’s our response to main points raised by the reviewer.
>
> >The justification for analyzing the symmetry of the data-model-algorithm system seems unclear. This is my major concern about this paper. ...
>
> Thanks for bringing this up. We have expanded our discussion to better motivate the importance of symmetries in the machine learning system. Roughly speaking, we need the symmetries induced by the system (learning algorithm) to be compatible with the learning task. Otherwise, we are trying to solve a task using wrong tools, which is highly inefficient. To be concrete, let's consider two examples and explore the role of symmetries.
>
> ***Example 1.*** Solving image classification using stationary kernels. Is this a good idea?  No! Because the kernel (and therefore the resulting (D, M, I) system) is rotationally invariant, if you rotate all images (including test images) by the same but arbitrarily chosen rotation ($O(3d)$), the test predictions are unchanged. We know such rotations completely destroy spatial information and the images are no longer `valid` (to humans). Therefore the symmetries of this system are not compatible with the data modality/learning task. Similarly, fully-connected networks are also rotationally invariant, so, for the same reason, we shouldn't use them for image classification! A consequence of this rotational invariance is that any mathematical theory based purely on deep fully-connected networks cannot fully explain the success of deep learning in image recognition --- if it could, then it would also predict good performance on `invalid`-image classification, while such “invalid”-images are completely unrecognizable to humans. Unfortunately, the vast majority of the deep learning theory papers focus on deep fully-connected networks.
>
> ***Example 2.*** On the other hand, suppose we want to solve a supervised learning problem in which the label is generated from a stationary Gaussian process. Is it a good idea to use convolutional networks? No! The task itself is invariant to any fixed orthogonal rotation on the input space, but the system induced by convolutional networks has a much more restricted symmetry group and is not invariant to this type of rotation. As a result, it would be an ineffective model, as its functional prior does not match the data distribution. Are fully-connected networks much better? Yes, if initialized with iid Gaussian weights (or any orthogonally-invariant initialization) since we know the induced system is then rotationally invariant. How about generic non-Gaussian (e.g. uniform) initialization? Perhaps not, because the system induced by the latter is only permutation-invariant and is still not completely compatible with the underlying symmetry of the task.
>
> To sum up, it is not about whether one symmetry group is better than another, it is more about whether the symmetry group is compatible with (or intrinsic to) the task. For more about the importance of symmetries in machine learning, see M. Bernstein’s excellent talk https://www.youtube.com/watch?v=w6Pw4MOzMuo and the book https://arxiv.org/abs/2104.13478.
>
> Now let’s come back to the fundamental question about which symmetry group is intrinsic to/compatible with the task of image classification. Human beings are certainly more comfortable with the identity group (no rotation) plus small variation. How about machines? As the reviewer pointed out, if we rotate each pixel along the channel dimension and allow pixels in different spatial locations to have independent rotations (i.e. $O(3)^d$ rotation in our paper), then the images are no longer valid (to humans!). Should we expect machines to perform badly? Very surprisingly, with such `invalid` images (for both training and test sets), EfficientNet B7 (first panel in figure4) does almost as well as with `valid` images (the top-1 accuracy on ImageNet: $\sim $ 83% vs $\sim$ 84%). That is to say, these `invalid` images are indeed valid to machines. This indicates that the mechanism used by machines to solve image classification tasks is quite different from that of the human vision system, and that perhaps the concept of “compatibility”/ “intrinsicity” of a symmetry group to tasks is different for humans and machines.
>
> >Studying the data-model-algorithm system as a whole seems to make the analysis in this paper more complicated than directly analyzing the model structure or the data...
>
> The proposal to focus on data-model-algorithms as an integrated system is one important contribution of our paper (though certainly not the only one!). While we agree with the reviewer that this perspective introduces additional complexity, this complexity is in fact necessary in order to adequately describe the behaviors and phenomena of real deep learning systems. Indeed, we have established empirical evidence that ignoring/modifying one component of this triplet dramatically alters the behavior of the system as a whole. We did an ablation study in Fig 1 by changing one variable at a time (changing one of D, M, or I), in which performance drops dramatically. Therefore, the system can no longer perform properly if one of (D, M, I) is modified in an incompatible way and we must treat the triplet (D, M, I) as an integrated system. As a specific example, we can not understand how deep learning works in image classification if we only analyze “deficient”  (D, M, I) system, e.g. when (1) D=natural images, M=Fully-connected networks, or (2) D=rotated images (O(3d)), M=ConvNet or (3) D=natural images, M=ConvNets, I=kernel regression. There is nontrivial interaction between all three components, and important phenomena/improved performance only occur with D=natural images, M=ConvNets, I=SGD and possibly some variation (e.g ConvNets-> ResNet, or even ViT, SGD-> Adam, etc.). Of course, in reality, (D, M, I) are often fixed and predetermined. This is because humans have spent more than half a century of efforts to find such systems and fine-tune them to optimality, from perceptrons (1958), to LeNet (1989), AlexNet (2012) and to ViT(2020).
>
> In sum, although analyzing the components of a system separately is simpler, it is highly unlikely that it can lead us to a complete picture of the mechanism behind the success of deep learning. We need to understand how (D, M, I) operates as an integrated system beyond what can be explained by each of its components separately.
> We hope this resolves the reviewer's concern about the necessity of considering a more complicated object than individual components and why we change one variable at a time.
>
> >The authors computed the eigen-decomposition of different models' one-hidden-layer infinite-width kernels on a particular input space and validated their results by experiments, but this setting might be too specific, making the results somewhat restricted.
>
> We disagree with the reviewer about this point. First, using spherical-type data (including high-dimensional isotropic Guassian) as input space is very standard in theory of deep learning. e.g. almost all double descent papers use isotropic Gaussian, see e.g. https://arxiv.org/abs/1908.05355. We are not aware of any prior work that can provide a precise characterization of the eigenspaces/eigenvalues in a very general setting besides spherical/gaussian/boolean cube type of data. Our theorem provides a very precise characterization of the eigen-structure of the kernel, including the dimensions, the eigenvalues, the unit eigenvectors of each eigenspace. This theorem serves as a base for rigorous theoretical analysis of data/training-efficiency of GAP/VEC/FCN, which will be included in a new version.  Alternatively, one could work on a very general space (e.g. arbitrary compact set) using Mercer's theorem. The analysis in this setting is indeed much simpler since we don't need to (and indeed cannot) characterize the eigenvalues, eigen-vectors, etc. As a trade-off, we lose the fine-grained information about the eigen-structure, which is necessary for furthering theoretical analysis. In addition, it is also very reasonable to begin with a one-hidden layer case. Indeed, there are hundreds of theoretical papers focusing on one-hidden layer networks, see e.g. https://arxiv.org/abs/1810.02054 about convergence and https://arxiv.org/abs/2101.10588 about generalization. Our 1-hidden layer result already provides novel and interesting insights about locality and shifting invariance  as mentioned by the first reviewer, distinguishing GAP from VEC, and VEC from FCN quantitatively. Multiple-layer convolution is much more complicated and requires new insights. It will be treated independently in future work.
>
> We will incorporate the suggestion from ``clarity`` part and ``Minor Comments`` part to further improve the paper.
>
> We hope the reviewer could consider increasing the score to acceptance if we have addressed the main concerns. If there are any remaining deficiencies, please let us know and we are happy to discuss them in more details.

---

> ### Author Response · Authors · 2021-08-23
> **Response to reviewer's Update**
>
> We thank the reviewer for the comments, and we are glad to be in agreement about the importance of symmetries in understanding convolutional networks.
>
> We acknowledge that the amount of the material presented in this paper exceeds what is typically found at NeurIPS. We could have broken this paper apart for multiple submissions, but in our view, a high-content paper such as this one has many advantages: it avoids incrementalism, it provides a strong single reference point for the ideas, and, most importantly, it allows for the telling of a complete story. We sincerely hope that there is still room for this type of work at NeurIPS.
>
> Of course, there are some disadvantages to such a high-content paper: it can be challenging to summarize even the main points in just 9 pages, and the overall narrative can end up being complex. As such, we agree with the reviewer that the presentation should be concise and focused to help the reader digest the material. For this reason, we have significantly streamlined the arguments and focused the discussion, especially the introduction (but throughout the paper as well). While we do not consider these revisions to be major (especially compared to the full scope of the paper), we do believe they address the reviewer’s concerns about Theorem 4.1/DIDE/experiments/prior work.
>
> It is unfortunate that we cannot share the revised draft to showcase these improvements. Instead, all we can do is offer the following Olympics analogy. It seems the reviewer’s main concerns are about “execution”. While we strongly believe the execution is more than satisfactory, we would also argue that this paper earns its top marks through ``difficulty”. A minor stumble on an easy floor routine might sink a gymnast’s medal chances, but the same stumble on a difficult routine would not keep her off the podium. We kindly ask the reviewer to bear this in mind when forming an overall judgment.

---

### Official Review · Reviewer_cNTz · 2021-07-16

**Rating:** 6
**Confidence:** 3

**Summary:**

The paper proposed an viewpoint to analyze deep learning methods as an integrated system of data (D), models (M), and inference algorithms (I) by studying the symmetric of the triplet (D, M, I) (in terms of transformations) on specific network models for vision including fully/locally connected networks (F/LCN), convolutional network w/o pooling(VEC/GAP). The key result in symmetry is that when networks are initialized with i.i.d. standard Gaussian distribution, then the considered neural networks are invariant against orthogonal group on the flatten input space when networks are over-parameterized.

**Ethical Concerns:**

No ethical issues with this paper.

**Limitations And Societal Impact:**

No potential negative societal impact of this work.

**Main Review:**

1. In the DIDE experiments for $VEC_n$, the authors claim that in Figure 3, the learning curve is dramatically speedup in the larger data set regime (# of samples $> 2^{12}$). However, the orange curve does not seem to have dramatic speedup from $2^{11}$ to $2^{15}$ comparing to the rest of the learning models. The authors then claim that it is because the model prior is too large (since $GAP_n \subseteq VEC_n \subseteq LCN_n \subseteq VEC_{dn}$ in (4)). I am not sure if Theorem 2.1 is able to interpret this empirical results. A more thorough discussion is needed.

2. The explanations on the experiment results in Figure 4 (middle) are not clear to me. Do you mean when the images are rotated, it is harder to learn but when as the number of the training samples increases, its become easier to learn (due to DIDE)? Why the red dots are not aligned with the orange dashed line then? How do you make sure the transition of the red dots is entirely due to DIDE? An ablation study is needed. It is possible to understand the benefit of DIDE in a more theoretical manner?

3. The explanation on Theorem 4.1 is also not clear to me. Why there are drops of eigenvalues in Figure 5? Can these drop be explained by Theorem 4? How does the proposed analysis on pooling connect to some existing works on analysis of pooling, e.g., [1]?

[1] Lee, C.Y., Gallagher, P.W. and Tu, Z., 2016, May. Generalizing pooling functions in convolutional neural networks: Mixed, gated, and tree. In Artificial intelligence and statistics (pp. 464-472). PMLR.

**Time Spent Reviewing:**

2.5

---

> ### Author Response · Authors · 2021-08-10
> **Response to Reviewer cNTz**
>
> We thank the reviewer for their time and constructive feedback. Here’s our response to specific points raised by the reviewer.
>
> 1.
>
> >In the DIDE experiments for VECn, the authors claim that in Figure 3, the learning curve is dramatically speedup in the larger data set regime (# of samples >212). However, the orange curve does not seem to have dramatic speedup from 211 to 215. comparing to the rest of the learning models.
>
>  Thanks for bringing up this point. We should have made it more clear that, by speeding up, we are comparing the *slope* in the larger dataset regime (e.g. $> 2^{12}$ VEC_n) v.s. that in the smaller dataset regime ($<2^{12}$ for VEC_n) of the *same* model, rather than across different models. This observation is in stark contrast to the classical setting where the slope is constant and independent of the dataset size; see also (Fig 1) in the well-known scaling law paper https://arxiv.org/pdf/2001.08361.
> Regarding
>
> > The authors then claim that it is because the model prior is too large (since
>  in (4)). I am not sure if Theorem 2.1 is able to interpret this empirical results. A more thorough discussion is needed.
>
> The phenomena of DIDE is new and we believe the theoretical understanding of it could provide valuable insights into the mechanism behind the success of deep learning. We totally agree Thm 2.1 is not sufficient to justify DIDE as it hasn't taken into account the interaction among data, model and optimization. We did not claim that Thm 2.1 is sufficient in our paper. Instead, we argue the coupled interaction among (D, M, I) is the main effect and we pose the theoretical understanding of this couple interaction and thus DIDE as an open question in the discussion section. Please also see the discussion [***Conjecture/Ideas of DIDE.***]  in the response to reviewer V4rf for a possible conjecture of DIDE.
>
>
> 2.
>
> > The explanations on the experiment results in Figure 4 (middle) are not clear to me. Do you mean when the images are rotated, it is harder to learn but when as the number of the training samples increases, it's become easier to learn (due to DIDE)? Why the red dots are not aligned with the orange dashed line then?
>
> Yes, exactly. As the number training samples increases, the learning curve becomes more efficient (slope increases) and eventually become as efficient as the unrotated one (i.e. two curves are almost parallel). This means, in terms of the efficiency of using extra data, the models with rotated and unrotated data are almost the same. However, we do not expect the two curves to collapse with each other in the log-log plot, unless the orange curve will plateau.
>
> >How do you make sure the transition of the red dots is entirely due to DIDE? An ablation study is needed.
>
> The only difference in the training configurations among the red dots are the training set sizes. Note that, for each training set size, we average over 3 runs (different random initialization) and plot both the mean and standard deviation. Since the only difference is the sample size, if it is not due to the increment of sample size, what else could it be? We also generate similar plots for different architectures, e.g. MLP-Mixer (right panel in Fig4), ResNet34/101 (fig S6) in the appendix and the same observations persist.
>
> Figure 3 is indeed an ablation study of DIDE, in which we add intervention to the triplet (D, M, I) by changing one variable at a time. E.g. we change $VEC_n$ to $VEC_\infty$ (an approximation of changing inference methods from SGD to kernel regression), change $VEC_n$ to $LCN_n$ (removing weight sharing), to $GAP_n/GAP_\infty$ (adding translation invariance, better prior). We also add rotation to the images (i.e. changing the D in (D, M, I)). With any of such intervention, the DIDE phenomenon disappears (or at least becomes much less obvious) and we do not observe a dramatic speed up as that in the $VEC_n$ case.
>
> >It is possible to understand the benefit of DIDE in a more theoretical manner?
> Please see the [***Conjecture/idea of DIDE***] in the response to reviewer V4rf, in which we layout a possible conjecture and a possible solvable mathematic model. Overall, this is an important, interesting but might be challenging theoretical question, requires us to model the rich interactions between (D, M, I) *beyond* the NTK/kernel regime, in particular the dynamics of how the *prior* are corrected. This question might also be very valuable to advance our understanding in deep learning and is currently out of the scope of this paper. We hope to address it in the future.
>
> 3.
>
> >The explanation on Theorem 4.1 is also not clear to me. Why there are drops of eigenvalues in Figure 5? Can these drop be explained by Theorem 4? How does the proposed analysis on pooling connect to some existing works on analysis of pooling, e.g., [1]?
>
> Unfortunately, due to the space constraint, we are not able to include a thorough discussion of this theorem. In the future version, we will explain and prove several theoretical consequences implied by the theorem, e.g. the polynomial improvement in both data and training efficiency due to locality.
>
> Regarding the drop of eigenvalues. Indeed, Theorem 4 precisely describes how the eigenvalues drop for various architectures and explains some puzzling phenomena in practice (e.g. why GAP architectures are much harder to train than VEC/FCN architectures). Roughly speaking, the mean trace of the kernel is $O(1)$. If an eigen-space is of dimension $N$, then the corresponding eigenvalue is at most $1/N$ (up to a constant factor). This is why we see a drop of eigenvalues (with multiplicity). The GAP drops faster because the dimensions of all eigen-spaces are much smaller (due to translation invariance and locality). As a result, with the same number of samples, the GAP kernel is able to cover eigen-spaces of much higher order, whose eigenvalues are much smaller (decay exponentially with the order). We will also add discussion to other existing work regarding pooling in future version.
>
> We hope we have addressed the reviewer’s main concerns and we would appreciate it if the reviewer could consider increasing their score.

---

### Official Review · Reviewer_V4rf · 2021-07-18

**Rating:** 8
**Confidence:** 4

**Summary:**

The paper studies the implications of various architectural choices on the performance of neural networks, and how they may break the curse of dimensionality. The authors consider fully-connected, locally-connected, locally-connected + weight sharing, and locally connected + global average pooling architectures. They then show, both theoretically and experimentally, how symmetries in the data may affect these architectures differently, at infinite width or finite width (where weight sharing can play a bigger role) under standard initializations. The authors also observe how the learning rates / slopes of learning curves may improve when the amount of data is large enough, particularly for finite networks. Finally, the authors provide spherical harmonic decompositions of the kernels that arise from infinite-width networks with different architectures -- fully-connected, locally connected, or locally connected with global pooling --, observing that local connectivity provides exponential (in the dimension) reductions in the size of eigen spaces, while pooling provides an additional improvement factor of the number of patches.

**Limitations And Societal Impact:**

yes

**Main Review:**

The question of what allows deep learning to learn efficiently by breaking the curse of dimensionality is important for understanding its success, and the paper provides an interesting and comprehensive study of this with both theoretical and empirical findings. Thus, I think the quality and significance of the paper are quite clear, and it is well written, thus I recommend acceptance.

Below are a few minor comments:
- When comparing LCN to VEC, the phrase "the opposite is also true" is a bit confusing, since this actually requires more parameters
- The setting of Figure 1 was a bit difficult to grasp and could be better explained (in the text or caption). If I understand correctly, the same spurious transformation is applied to each example? Also, the y axis lacks a label.
- For DIDE, do you have an idea/conjecture of what additional "priors" are being learnt by the networks at large data regimes for improving these slopes?
- It would be helpful to add a paragraph after Theorem 4.1 explaining how the improvements in dimensionality/spectral gaps may improve learning efficiency (instead of simply pointing to the appendix)
- Note that similar questions regarding benefits of locality and/or pooling in kernel regimes have been studied in a few recent papers, e.g., [here](https://arxiv.org/abs/2102.13219), [here](https://arxiv.org/abs/2102.10032), [here](https://arxiv.org/abs/2106.08619). It would be good to provide comparisons at least in the final version of the paper.
- typos: L74 "even" -> "even when"?; L163 "Kernelss"; L277 "unrorated"; L305 "harmonic" -> "harmonics"


-------- update after rebuttal ----------
Thanks for your response. The discussion raised some concerns regarding clarity, and I hope the authors can address these (see comments by reviewer XBfK). Nevertheless, I think these are minor issues which can be addressed for a final version, and the authors' last response to reviewer XBfK makes me confident that they will be properly addressed, thus I decided to raise my score.

**Time Spent Reviewing:**

3

---

> ### Author Response · Authors · 2021-08-10
> **Response to Reviewer V4rf**
>
> We thank the reviewer for their time and constructive feedback. Here’s our response to specific points raised by the reviewer.
>
> >When comparing LCN to VEC, the phrase "the opposite is also true" is a bit confusing, since this actually requires more parameters
>
> Thanks for this point. We will clarify it in a newer version.
>
> >The setting of Figure 1 was a bit difficult to grasp and could be better explained (in the text or caption). If I understand correctly, the same spurious transformation is applied to each example? Also, the y axis lacks a label.
>
> Yes, the *same* transformation is applied to each image, which is equivalent to applying the adjoint transform to the coordinate system of the input space. We will add more detail to better explain the results. The y-axis is accurate and will add the label.
>
> >For DIDE, do you have an idea/conjecture of what additional "priors" are being learnt by the networks at large data regimes for improving these slopes?
>
> [***Conjecture/Ideas of DIDE.***] This is a great question indeed and we appreciate the reviewer's interest in it. For concreteness, let's consider the task of learning a translation invariant harmonic polynomial of degree $r$  using VEC as an example, in which an ideal prior is GAP. We assume this polynomial is a sum of $p$ components and each component depends on a patch of size $d_0$.  The ideal prior GAP takes about $d_0^r$ many samples (optimal in generic setting). Let us explain why finite width VEC could almost achieve such optimal sample-efficiency while infinite VEC kernel need at least $pd_0^r$ many samples, where $p$ is the number of patches, $d_0$ is its size and $pd_0=d$. Now for finite width VEC (with weight sharing), learning the linear part will require $\sim d$ examples. Note that the linear part is translation invariance, and we conjecture the *pooling* (translation invariance, the group is of order $p$) is approximately learned.  When moving to learn higher order terms, we already *learned* this symmetry in the representation and does not need to *re-learn* it (which is the case in kernel), saving us a factor of $p$ in terms of data-complexity. Therefore, the number of data points required is approximately $max(d, d_0^r)$ rather than $pd_0^r$ for learning such polynomials. That is to say, finite-width VEC only needs to learn the symmetry *once* and this symmetry can *propagate* to all higher frequencies while the corresponding kernel requires learning it $r$ many times, i.e. the learned symmetry does not propagate from one frequency to another. If you look at figure S5 in appendix, 4th row and 5th column,  you will see the finite width VEC learns the linear part slowly (when $log(m)/log(d)\sim 1.4$) and then rapidly learns degree 2 & 4 parts (when $log(m)/log(d)\sim 2$). We see the efficiency of finite width networks are noticeably better than that of kernels in this setting. However, when we swap out $VEC_n$ with $LCN_n$ (no weight-sharing) sample-efficiency of kernels and of finite width counterparts are almost identical, see third row last column in S5. One explanation for this is that, without (learning the) weight-sharing, the network is not translation invariance even if the pooling layer is learned. Overall, there is some interesting mathematics even in this particular example that is worth digging into.
>
> >It would be helpful to add a paragraph after Theorem 4.1 explaining how the improvements in dimensionality/spectral gaps may improve learning efficiency (instead of simply pointing to the appendix)
>
> Thanks for the suggestion. In the new version, we will have a more thorough discussion about the implication of this theorem to both the learning and training efficiencies in the main text. We will explain and prove quantitative improvement in both data and training efficiency due to locality/symmetries.
>
> >Note that similar questions regarding benefits of locality and/or pooling in kernel regimes have been studied in a few recent papers, e.g., here, here, here. It would be good to provide comparisons at least in the final version of the paper.
>
> Thanks for the references. We will add citations of these excellent works to our final version and provide comparisons. Note that the last paper in the reference list "Locality defeats the curse of dimensionality in convolutional teacher-student scenarios", concurrent to ours (appeared on arXiv on June 16th which is after the submission of our paper to NeurIPS), makes similar observations/conclusions to that of section 4 in our paper. However, their approach is slightly different from ours: they use Mercer’s theorem while we use spherical harmonics. We will include comparisons in the final version.
>
> >Thanks for correcting our typos!

---

> ### Author Response · Authors · 2021-08-25
> **update**
>
> Thanks for increasing the score and we sincerely appreciate your trust in us to improve the clarity of the paper!

---

### Author Response · Authors · 2021-08-24
**Message to AC and Reviewers**

Dear AC and Reviewers,

We understand that there are a lot of efforts spent behind-the-scenes, in particular given the large number of papers assigned to each AC and each reviewer this year. If there is something (clarification, further questions, etc) we could help to ease the review process of this paper, please don't hesitate to reach out and we are more than happy to answer the questions.

Best regards,
Authors

---

### Decision · Program_Chairs · 2021-09-27

**Decision:**

Reject

**Comment:**

The paper studies how symmetries and biases associated to various architectural choices may alleviate the curse of dimensionality. The reviewers found the methodology employed by the authors interesting, and generally appreciated the various observations and experiments provided in the paper. That said, the reviewers raised several critical concerns pertaining to presentation (experimental details, in particular), clarity, and focus. Despite the considerable discussion this paper has generated among the reviewers, opinions remained divided whether a substantial revision is needed before the paper can be considered for publication.